

# Improving algorithms and uncertainty estimates for satellite NO₂ retrievals: Results from the Quality Assurance for Essential Climate Variables (QA4ECV) project

K. Folkert Boersma[1,2], Henk J. Eskes[1], Andreas Richter[3], Isabelle De Smedt[4], Alba Lorente[2], Steffen Beirle[5], Jos H. G. M. van Geffen[1], Marina Zara[1], Enno Peters[3], Michel Van Roozendael[4], Thomas Wagner[5], Joannes D. Maasakkers[6], Ronald J. van der A[1], Joanne Nightingale[7], Anne De Rudder[4], Hitoshi Irie[8], Gaia Pinardi[4], Jean-Christopher Lambert[4], and Steven C Compernolle[4]

[1]Royal Netherlands Meteorological Institute, Satellite Observations department, The Netherlands

[2]Wageningen University, Meteorology and Air Quality Group, Wageningen, the Netherlands

[3]Institute of Environmental Physics (IUP-UB), University of Bremen, Bremen, Germany

[4]Belgian Institute for Space Aeronomy (BIRA-IASB), Brussels, Belgium

[5]Max-Planck Institute for Chemistry (MPI-C), Mainz, Germany

[6]Harvard University, Cambridge, Massachusetts, United States

[7]National Physics Laboratory (NPL), Teddington, United Kingdom

[8]Center for Environmental Remote Sensing (CEReS), Chiba University, Chiba, Japan

*Correspondence to*: K. Folkert Boersma (boersma@knmi.nl)

**Abstract.** Global observations of tropospheric nitrogen dioxide (NO₂) columns have been shown to be feasible from space, but consistent multi-sensor records do not yet exist, nor are they covered by planned activities on the international level. Harmonised, multi-decadal records of NO₂ columns and their associated uncertainties can provide crucial information how the emissions and concentrations of

nitrogen oxides evolve over time. Here we describe the development of a new, community best practice NO₂ retrieval algorithm based on a synthesis of existing approaches. Detailed comparisons of these approaches led us to implement an enhanced spectral fitting method for NO₂, a 1°×1° TM5-MP data



assimilation scheme to estimate the stratospheric background, and improve air mass factor calculations. Guided by the needs expressed by data users, producers, and WMO GCOS guidelines, we incorporated detailed per-pixel uncertainty information in the data product, along with easily traceable information on the relevant quality aspects of the retrieval. We applied the improved QA4ECV $NO_2$ algorithm on

the most actual level-1 data sets to produce a complete 22-year data record that includes GOME (1995-2003), SCIAMACHY (2002-2012), GOME-2(A) (2007 onwards) and OMI (2004 onwards). The QA4ECV $NO_2$ spectral fitting recommendations and TM5-MP stratospheric column and air mass factor approach are currently also applied to S5P-TROPOMI. The uncertainties in the QA4ECV tropospheric $NO_2$ columns amount to typically 40% over polluted scenes. First validation results of the QA4ECV

OMI $NO_2$ columns and their uncertainties over Tai'an, China in June 2006 suggests little bias (-2%) and better precision than suggested by uncertainty propagation. We conclude that our improved QA4ECV $NO_2$ long-term data record is providing valuable information to quantitatively constrain emissions, deposition, and trends in nitrogen oxides on a global scale.

# 1 Introduction

Nitrogen oxides ($NO_x$ = NO + $NO_2$) in the atmosphere have far-reaching effects on the Earth system. In the lower troposphere, nitrogen oxides promote the photochemical production of ozone (e.g. Liu et al. [1987]; Grewe et al. [2010]), whereas in the stratosphere, $NO_x$ leads to the catalytic destruction of ozone, and the formation of reservoir species for halogens (e.g. Crutzen et al. [1970]). Nitrogen oxides

contribute to aerosol formation, and they are linked to the oxidizing efficiency of the troposphere via ozone which plays an important role in the formation of the hydroxyl radical (OH). $NO_2$ itself is only a weak greenhouse gas [Solomon et al., 1999], but has considerable relevance for radiative forcing because nitrogen oxides are important precursors of tropospheric ozone, aerosols, and OH. The net effect of nitrogen oxides on climate forcing is modeled to be negative, or 'cooling', with $NO_x$-driven

aerosol screening dominating over tropospheric ozone warming [Shindell et al., 2009]. In 2011, the World Meteorological Organization (WMO) Global Climate Observing System (GCOS) included $NO_2$ (together with $SO_2$, HCHO, and CO) in its Implementation Plan for the Global Observing System for



Climate in Support of the UNFCCC [WMO, 2011] "in recognition of the emission-based view on climate forcing of ozone and secondary aerosols, relevant for climate mitigation and important for processes". The formal attribution of $NO_2$ as precursor to the ozone and aerosols Essential Climate Variables, or ECVs [Bojinski et al., 2014], implies that the scientific community has committed itself to

providing reliable, long-term measurement records of $NO_2$. Apart from its relevance to climate change, atmospheric nitrogen oxides are also important for the health of ecosystems and humans. Deposition of nitrogen to ecosystems may affect the structure and functioning of ecosystems (e.g. Galloway et al. [2003]). Recently, the World Health Organization stated that it is reasonable to infer that $NO_2$ has direct short-term health effects, such as airway inflammation and reductions in lung function [WHO, 2013],

and a literature review of epidemiological studies over a wide geographic area by Hoek et al. [2013] showed that human mortality was significantly associated with long-term exposure to $NO_2$.

High-quality observations are needed to monitor the concentrations of nitrogen oxides in the atmosphere, both close to the ground, where $NO_2$ is relevant for deposition and health aspects, as well as aloft, where nitrogen oxides influence atmospheric chemistry and climate. Such measurements are

useful for reanalysis studies (e.g. Inness et al. [2013]), contribute to documenting changes in $NO_2$ concentrations and $NO_x$ emissions (e.g. Zhang et al., [2008]; Vinken et al. [2014]), and to attributing any such changes to their underlying causes (e.g. Verstraeten et al., [2015]; Xu et al., [2013]). This provides policy makers with options for decisions to counter environmental problems (e.g. Witman et al. [2014]). Measurements may also enhance the public's appreciation of the extent and scope of the

problem of air pollution. In situ measurements of $NO_x$ concentrations taken on the ground are representative for the quality of the air people breathe close to the measurement station. But such stations are relatively scarce in many countries and cannot provide a global and spatially continuous perspective. Satellite observations on the other hand provide global coverage, thereby offering the unique opportunity to study spatial patterns and temporal variation in $NO_2$ pollution. For any type of

measurement, it holds that they can only be used properly in science or as evidence-basis for policy decisions, if there is unequivocal confidence in the data sets, as well as a proper understanding of their limitations.



The EU Seventh Framework (FP7) project Quality Assurance for Essential Climate Variables (QA4ECV [2018], www.qa4ecv.eu) was designed to demonstrate how reliable climate data sets can be generated, along with detailed and traceable information on the quality of such data. Specifically, for $NO_2$, the goals of this project are: (1) to generate a multi-decadal (1995-2017) satellite data record of

tropospheric and stratospheric $NO_2$ column densities based on calibrated satellite data and state-of-the-art retrievals, and (2) to provide fully traceable uncertainty metrics for this record, ready for ingestion in models or in other interpretation efforts. Obtaining global, long-term, and stable satellite observations with validated accuracy and precision is not straightforward. The GOME (1995-2003; Burrows et al. [1999]), SCIAMACHY (2002-2012; Bovensmann et al. [1999]), OMI (from 2004 onwards; Levelt et al.

[2006]), and GOME-2A (from 2007 onwards; Munro et al. [2007]) instruments have been providing global observations of $NO_2$ over the last 22 years, but there are important differences in overpass time, instrumental artefacts (e.g. calibration and design differences), and signal-to-noise levels that need to be taken into account. To be used properly, the information content of the $NO_2$ products needs to be validated over a variety of regions, and users need guidance provided by well-established quality

information to help them judge the fitness-for-purpose of the $NO_2$ products.

In this work, we demonstrate our approach to improve a retrieval algorithm and apply it to generate a multi-decadal record of $NO_2$ columns with a consortium of European retrieval groups. We follow the guidelines for the generation of ECV datasets from WMO [2010]. Our efforts are inspired by the QA4ECV project goals described above, but also by recent studies showing that there is still room for

substantial improvement in all sub-steps of the retrieval (e.g. Richter et al. [2011]; Lin et al. [2014]; van Geffen et al. [2015]; Krotkov et al. [2016]), by the outcome of validation studies showing that various state-of-science retrievals have biases on the order of tens of percents (e.g. Jin et al. [2016]; Drosoglu et al. [2017]; Kollonige et al. [2018]), and the considerable structural uncertainty in retrieved tropospheric $NO_2$ columns emerging when different retrieval methodologies are applied to the exact same satellite

observations (e.g. van Noije et al. [2006]; Lorente et al. [2017]). The efforts from five European retrieval groups within the QA4ECV consortium allow us to perform a detailed comparison of current approaches to various retrieval sub-steps. These comparisons have proven to be helpful in reducing and



better quantifying the uncertainty of the $NO_2$ retrieval. The improved quality of the QA4ECV $NO_2$ record itself, and the improved knowledge on the uncertainties, should make the QA4ECV satellite data record better fit for the purpose of trend analysis, data assimilation, and inverse modelling studies.

The manuscript is organised as follows: in section 2 we discuss how $NO_2$ data user requirements, the
expertise from $NO_2$ data providers, and the quality requirements defined by GCOS are providing direction for this study. In section 3 we assess the quality of the best currently available level 1 data sets for $NO_2$ retrieval from GOME, SCIAMACHY, OMI, and GOME-2(A), and discuss how this guides the selection of spectral fitting approaches. Section 4 focuses on the algorithm design and the traceability of the retrieval approach and external data used. In section 5, we give an overview of the main lessons
learned in the inter-comparisons of retrieval sub-steps. Section 6 summarizes the uncertainty information provided in the QA4ECV data product, and how these uncertainty estimates compare to the inter-comparison results from section 5. We conclude with a first validation of our new QA4ECV OMI $NO_2$ tropospheric columns and their uncertainties against independent MAX-DOAS measurements collected during a one-month campaign over Tai'an, China.

## 2 User needs and expert recommendations

### 2.1 User survey

At the start of the QA4ECV project, we identified the requirements of data users in terms of uncertainty information and usability of the data product. This included a survey of 22 $NO_2$ data users, and interviews with 3 $NO_2$ 'champion users', who provided more detailed written answers to questions. The
questionnaire was aimed at establishing what users need in terms of quality flags, traceability information, and product uncertainty description. Here we briefly summarize the main outcome of the survey for $NO_2$. The full survey also includes results for the HCHO and CO data products and can be found in QA4ECV Deliverable 1.1 [Nightingale et al., 2015].

The $NO_2$ users, mostly from academia and policy support, responded that $NO_2$ data products available

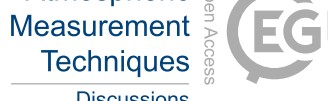

at the time generally provide quality flags, but that the information is limited. They recommended that the data products should include more detailed quality flags describing the condition of low quality measurements, along with a master flag ('use' or 'do not use') for quick inspection. Figure 1 indicates that $NO_2$ data users ask for information on whether scenes are affected by cloud and aerosol

5  contamination, sun glint, sensor status (e.g. row anomaly), and whether measurements were done over land or over sea, or over snow/ice.

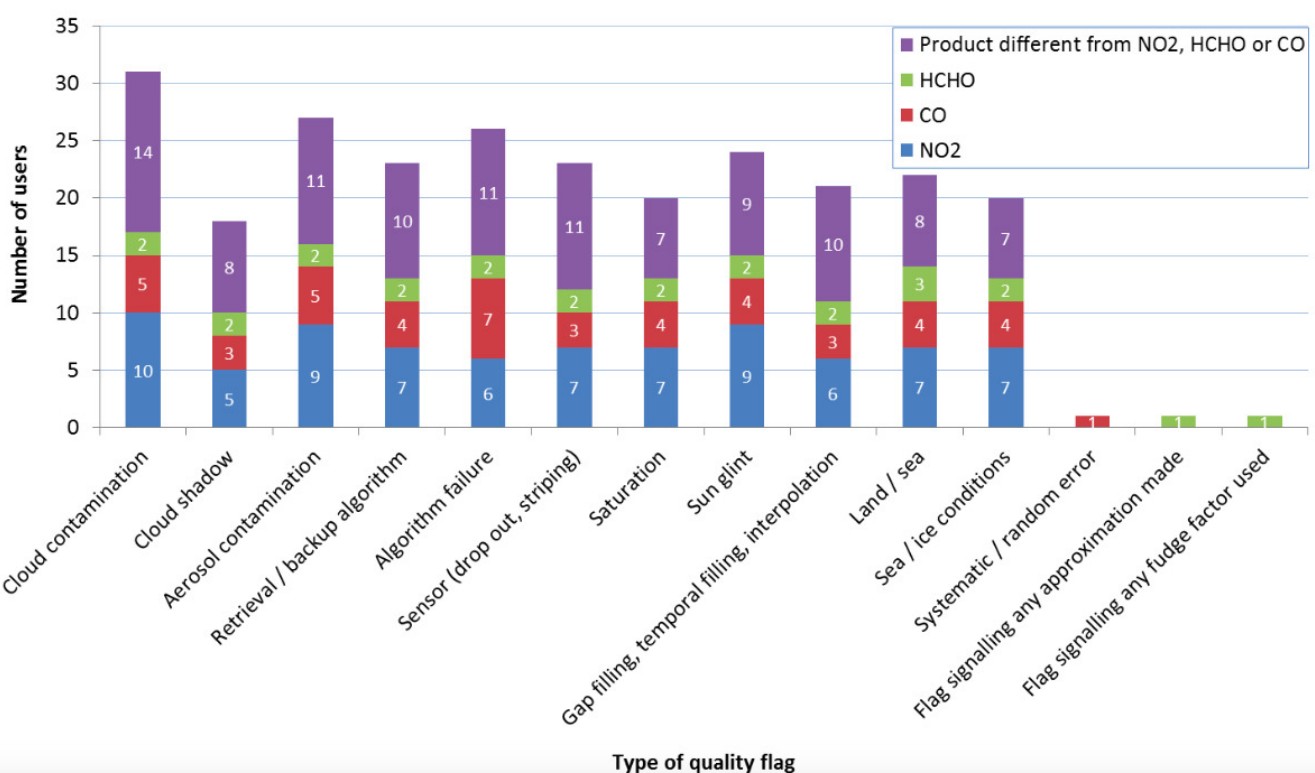

**Figure 1.** Response to the user survey question 'What additional information would you like to see provided as a quality flag?' in a $NO_2$ satellite data product (blue bars).

Current $NO_2$ data products often contain information on algorithm uncertainty on a per-pixel basis, but users indicated that they need specific information on the systematic and random error parts contributing to the stated uncertainties, on long-term stability of the data record, and on the dependence of the uncertainties on the ancillary parameters cloud fraction, cloud pressure, surface albedo, aerosols,

15  and temperature. Users said they needed this information in inverse modeling and data assimilation



experiments in order to apply realistic weights to the observations and the modeled fields, and for weighting and filtering in trend analyses and mapping.

Respondents also stated that it was important to provide traceability information and processing information along with the $NO_2$ data product. Full schematic details of a processing chain for the $NO_2$ satellite data product should be provided. When asked why they needed such information, the most common answers were: "to understand the data", "to identify sources of uncertainty" (in the retrieval algorithm), to apply appropriate "data filtering", and "to account for uncertainty in further processing" of the $NO_2$ data for their own purposes. Traceability information would have to be made available at a point of central access, along with other documentation on the data processing, such as an Algorithm Theoretical Baseline Document (ATBD) and a Product Specification Document (PSD) with guidance on how to use or not use the data.

Last but not least, the users found a systematic validation of the $NO_2$ data product with coherent independent reference data to be desirable, especially if the independent reference data itself is properly quality-assured. Information on the validation status of the product would preferably be gathered in a central access point, along with the traceability information. Within the QA4ECV project, these recommendations have led to the development of a so-called Quality Assurance (QA) System. Within this system (available at: http://www.qa4ecv.eu/qa-system), data producers have the possibility to provide all these pieces of information, and users can obtain a quick overview of the maturity and completeness of the data product.

## 2.2 Producer requirements

We also carried out a survey of data producer requirements for quality assurance in satellite data records, and discussed retrieval priorities and quality assurance (QA) needs with retrieval experts from different groups within the consortium (BIRA-IASB, IUP Bremen, KNMI, Max Planck Institute for Chemistry, and Wageningen University in alphabetical order).



Producers of data products (other than those involved in QA4ECV) that we interviewed recognized the need for processing chain information to be more transparent and more easily accessible for data users. They also admit that traceable input (ancillary data) files, read-in software, sensitivity analysis documentation, and publications are not always provided along with the data. Data producers stated that

direct communication with their data users is important, mostly on issues including read-in software, product format, flagging and filtering procedures, and the uncertainty budget. In general, data producers thought that the necessary traceability and quality information is in principle available, but cannot always be easily found. These recommendations helped shape QA4ECV QA System further. Within this system, data producers have the possibility to provide all these pieces of information, and users can

obtain a quick overview of the maturity and completeness of the data product. Data producers were generally positive about benchmarking their satellite data product against other scientific standards, such as from cross-calibrated global validation networks. They noticed that quality information for independent reference data is often not available. For more detailed outcomes of the ECV data producer survey, please see QA4ECV Deliverable 1.1 [Nightingale et al., 2015].

There was also a strong intrinsic motivation from $NO_2$ data producers to improve the retrieval algorithm and generate a long-term $NO_2$ data set from available satellite reflectance measurements. The $NO_2$ retrieval groups in the QA4ECV consortium discussed priorities for retrieval improvement, based on their collective experience with the retrieval, validation, and use of existing individual $NO_2$ data

products for different sensors. The central idea was to arrive at a QA4ECV 'consortium algorithm' based on best practices derived from lessons learned from inter-comparisons between approaches for all relevant retrieval sub-steps and extend the steps initiated within the ESA S5P verification project [DLR, 2015].

Retrieval of tropospheric $NO_2$ columns is based on a 3-step approach. First, a set of absorption cross-sections, including $NO_2$, is fitted to the measured top-of-atmosphere reflectance spectrum, which provides the so-called slant column densities (SCDs, $N_s$). Then (step 2), the stratospheric contribution to the SCD ($N_{s,strat}$) is estimated and subtracted from the SCD. In the third step, the tropospheric air mass



factor (or AMF, $M_{trop}$) is calculated based on knowledge of the satellite viewing conditions and assumptions on the state of the atmosphere in order to convert the residual tropospheric SCD into a tropospheric vertical column density VCD ($N_{v,trop}$). The retrieval equation is:

$$N_{v,trop} = \frac{N_S - N_{S,strat}}{M_{trop}}$$ (1)

The following activities leading to retrieval improvement were identified and conducted during the QA4ECV project:

(1) An inter-comparison of spectral fitting approaches between institutes. $NO_2$ SCDs were computed by all groups for the same orbits of level-1 data and results were compared. This

resulted in a quantification of the level of agreement on the slant columns and a better understanding of the factors responsible for remaining differences. This is a relevant exercise in view of the substantial revisions of spectral fitting approaches over the last years (e.g. Richter et al. [2011]; van Geffen et al. [2015]; Marchenko et al. [2015]; Anand et al. [2015]), and resulted in the definition of the QA4ECV 'best practices' spectral fitting algorithm.

(2) An evaluation of the algorithm SCD uncertainties against an independent statistical uncertainty estimates [Zara et al., 2018].

(3) A comparison of stratospheric $NO_2$ fields and associated tropospheric residues from different approaches: consistency and plausibility checks, and quantification of differences. Recent improvements in the KNMI data assimilation approach [Maasakkers, 2013], and the newly

developed STREAM scheme [Beirle et al., 2016], provided more insight in the stratospheric correction, and in the associated uncertainties.

(4) A comparison of altitude-dependent, or 'box' air mass factors (AMFs) for simplified scenarios. This comparison established the degree of consistency between radiative transfer models, pointed out discrepancies, and provided hints for possible improvements. The resulting spread

between the (box) AMFs can be interpreted as the structural uncertainty[1] when using different

---

[1]Structural uncertainty can be identified with the metrology concept "uncertainty of measurement method" (see the Guide to the expression of uncertainty in measurement [GUM, 2008], section F.2.5): *uncertainty associated with the method of*



radiative transfer models, vertical layering and interpolation schemes [Lorente et al., 2017].

(5) Comparisons of tropospheric AMFs calculated by different groups with an increasing number of differences in algorithm choices: from identical settings (wherein only model, vertical layering, and interpolation differ between groups), via preferred settings (every group using their own preferred information on clouds, albedo, $NO_2$ profile, etc.), to a wider 'round robin' comparison wherein also groups outside of Europe participated. This last comparison was 'unguided', i.e. groups could freely decide how to calculate their AMFs, deciding for themselves whether to include aerosol corrections, using look-up tables, correcting for residual clouds etc. The spread between the round robin AMFs is indicative of the structural uncertainty in the AMF calculation [Lorente et al., 2017].

It is impossible at the algorithm development stage to have a full understanding which settings and apporaches lead to the best results. This led the consortium to consider it beneficial to include more than one 'best practices' approach for the stratospheric correction and AMF calculation sub-steps. Specifically, apart from the proposed 'default' stratospheric correction method, also stratospheric $NO_2$ column estimates from the independent STREAM method has been included in the QA4ECV $NO_2$ data product. For the tropospheric AMF calculation, it was decided to provide both the standard tropospheric AMF (linear combination of a partly cloudy, partly clear-sky AMF) but also to include the clear-sky AMF in the data product. This allows data producers to directly test different 'retrieval options' (correcting for residual clouds vs. cloud 'clearing') at the validation stage, and provides users with the possibility to test the robustness of the data product beyond the quoted retrieval uncertainty alone.

## 2.3 GCOS requirements and GCOS guidelines for dataset generation

The Global Climate Observing System (GCOS) published a set of requirements that tropospheric $NO_2$ columns should fulfill. The requirements from GCOS report 154 [WMO, 2011] are listed in Table 1 below. The recently published requirements from GCOS report 200 [GCOS, 2016], are not considered

_measurement, as there can be other methods, some of them as yet unknown or in some way impractical, that would give systematically different results of apparently equal validity._



here yet.

**Table 1.** GCOS requirements for satellite retrievals of tropospheric $NO_2$ columns [WMO, 2011].

| | Horizontal Resolution | Vertical Resolution | Temporal Resolution | Uncertainty | Stability/decade[2] |
|---|---|---|---|---|---|
| $NO_2$ tropospheric column | 5-10km | N/A | 4h | max(20%; 0.03 DU[1]) | 2% |

[1]An uncertainty of 0.03 DU (Dobson Units) corresponds to $0.8 \times 10^{15}$ molec.cm$^{-2}$. The 0.03 DU holds for

tropospheric $NO_2$ columns up to $4.0 \times 10^{15}$ molec.cm$^{-2}$, for larger column values the relative uncertainty of 20% holds. Note that we replaced the heading 'accuracy' of [WMO, 2011] by 'uncertainty' to be compliant with ISO standard on metrology [VIM]. Indeed, [WMO, 2011] states that "the [accuracy] requirements are indicative of acceptable overall levels for the uncertainties of product values."

[2]Stability is in general understood to represent the extent to which the error of a product remains

constant over a long period, typically a decade or more [WMO, 2011].

The GCOS requirements, especially those on resolution, can be discussed for their adequacy. These are 'target requirements', which should be advanced towards when generating a long-term record of tropospheric $NO_2$ column measurements. The resolution requirements listed above cannot be met by the

satellite sensors capable of measuring $NO_2$ that have been operational over the last 20 years, because of limitations in their instrument design, with the exception of the recently launched S5P-TROPOMI sensor, which does meet the requirement. Indeed, the GCOS report states that "products at lower spatial and temporal resolution" than 5-10 km (that is the $NO_2$ products currently available from GOME, SCIAMACHY, OMI, and GOME-2) "…would be sufficient to provide an independent instrument data

record of long-term precursor trends to assist in the attribution of changes in ozone and aerosol".

The target requirements for uncertainty and stability are possibly within reach, judging from validation studies, and these have motivated the QA4ECV consortium to find ways to reduce the retrieval



uncertainties, and to better estimate the systematic error component of the retrieval uncertainty.

GCOS has also established guidelines for the generation of climate datasets [GCOS, 2010]. Those guidelines serve as a checklist, against which ECV producers can evaluate their production and

documentation process. Section 1 of the Supplement provides a point-by-point overview of how these guidelines have been taken into account for the generation of the QA4ECV $NO_2$ data product. A comprehensive comparison with respect to these and other GCOS requirements [GCOS, 2016; WMO, 2010, 2011] is available in QA4ECV deliverable D6.1 [Compernolle, 2018].

## 3 Quality of level 1 data

In the early stages of the QA4ECV project design, it was decided to use GOME, SCIAMACHY, OMI, and GOME-2 (on MetOp-A) to generate a data record for tropospheric and stratospheric $NO_2$ vertical columns spanning the period 1995-2017. Table 2 lists the relevant instrument specifics for these instruments. For all instruments, the most recent and corrected level-1 datasets are used.

**Table 2.** Satellite instruments and level-1 data contributing to the QA4ECV $NO_2$ ECV data product.

| Instrument | Local overpass time | Spatial resolution | Calibrated level-1 datasets | Spectral resolution /sampling | Main level-1 issue |
|---|---|---|---|---|---|
| GOME (1995-2003)[1] | 10:30 hrs | $320 \times 40$ km$^2$ | version 5 | 0.40 nm, 0.20 nm | Spectral structures in Solar irradiance caused by diffuser plate |
| SCIAMACHY (2002-2012) | 10:00 hrs | $60 \times 30$ km$^2$ | version 7.04-w | 0.44 nm, 0.24 nm | |
| OMI (2004-) | 13:40 hrs | $24 \times 13$ km$^2$ | collection003 | 0.63 nm, 0.21 nm | Row anomaly (blockage), stripes |
| GOME-2(A) (2007-) | 09:30 hrs | $80 \times 40$ km$^2$ | EUMETSAT/ R/5_12 | 0.50 nm, 0.20 nm | Throughput loss resulting in more noise |

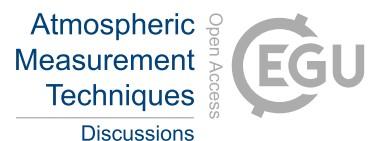

[1]GOME lv1 data is in principle available up until September 2011, but for a limited area of the globe only. In June 2003 the on-board tape recorder failed resulting in reduced coverage of GOME-observations, since data could only be downlinked in 'real-time' during overpasses above ground receiving stations.

Prior to algorithm testing, we assessed the quality of the relevant level-1 data. Here we briefly discuss our findings, and discuss how the quality of the level-1 data may affect the retrieval of $NO_2$ SCDs and their uncertainties.

**GOME**

GOME level-1 data with global coverage is available from July 1995 to June 2003. ESA has produced a GOME level-1 data set for the mission called version 5.1 [GOME Products and Algorithms, 2018] that is sufficiently well characterized and complete. An important concern with GOME level-1 data is that the solar irradiance signal is detected after reflection from a diffuser plate, whereas the radiance signal

is not. The reflection on the diffuser plate created large and seasonally varying artificial spectral structures in the solar irradiance [Richter and Wagner, 2001]. This makes it very difficult for GOME to use solar irradiance spectra as reference in the DOAS spectral fitting. To avoid the issue, Earthshine radiances over remote regions can be used as reference spectra. The implication is that only differential $NO_2$ SCDs are retrieved. To provide the total $NO_2$ SCDs necessary for an ECV dataset of stratospheric

and tropospheric $NO_2$ columns, a background correction, typically estimated from an external source, is required.

Detector degradation is another relevant issue for $NO_2$ and cloud retrievals in the visible channel. This degradation in the level-1 data has been estimated to amount to approximately 15% between 1995 and

2003 [Slijkhuis et al., 2015], and is anticipated to result in modest increases in the GOME $NO_2$ SCD uncertainty. The quality of the GOME level-1 data has also been affected by other instrument-related issues, but these may be of less relevance to the quality of the $NO_2$ spectral fits. Different GOME scan angles (East, Nadir, West) are affected differently in terms of throughput degradation and dichroic



mirror degradation, possibly resulting in systematic differences in $NO_2$ SCDs for the different scan angles (or stripes).

## SCIAMACHY

SCIAMACHY lv1 data is available from August 2002 until April 2012. SCIAMACHY lv1 version 7.04 data has been made available by ESA in 2016. One particular feature of the SCIAMACHY level 1 data is that co-adding of spectra was performed on-board of SCIAMACHY, prior to downlinking the data from the satellite to receiving stations. The cluster 424-527 nm was read out more frequently (than other spectral bands) in order to minimize the co-addition of spectra and thereby optimizing the spatial

resolution for $NO_2$ to $60 \times 30$ km$^2$ (30 x 30 km$^2$ in some latitude bands). A consequence of this is that only spectral data from the 424-527 nm cluster are available for DOAS $NO_2$ spectral fitting. Similar as for GOME, SCIAMACHY solar irradiances suffer from spectral structures from the diffuser plate. A second diffuser was therefore included in the instrument, mounted on the backside of the azimuthal scan mirror. Using solar irradiances from this azimuthal scan mirror strongly reduces the apparent

seasonality in $NO_2$ introduced by the diffuser, although some structures still remain [Richter et al., 2011].

Over its lifetime, the SCIAMACHY instrument suffered from degradation of its optical components. This degradation is the result of a complex mixture of aging of the front optics through UV radiation

and photochemical reactions, detector contamination by water vapour deposition and changes in the thermal equilibrium of the platform. As a result, the throughput of SCIAMACHY decreased over the years, in particular in the UV. In addition, small changes in spectral sensitivity over time, for example from etaloning[2], cancel when using daily irradiance spectra for DOAS spectral fits, but this prevents the use of a single solar irradiance for the full time series. As degradation of the scan mirror leads to scan

---

[2]Etaloning refers to unintended multiple reflection of radiation between optical elements leading to constructive and destructive interference for some wavelengths.



angle dependent degradation, and scan angle dependent biases, or stripes, can therefore develop over time in the NO$_2$ SCDs.

**OMI**

The OMI instrument performance and level-1 data product have proven stable (to within 2%) since the mission started in October 2004. The OMI level-1 data are from the Collection 3 data. Processing of this Collection 3 data started in February 2010 with the version 1.1.3 of the Ground Data Processing System software [Dobber et al., 2008], and has produced a complete level-1 dataset for the entire OMI mission. The main issue of the OMI level-1 data is the so-called row anomaly (RA). From June 2007 onwards,

several rows of the CCD detector (each corresponding to a specific part of the OMI nadir field-of-view), received less light from the Earth, and some other rows appear to receive light directly from the Sun. A plausible reason for these effects is a partial obscuration of the entrance port by insulating layer material that may have come loose on the outside of the instrument. For rows affected by the RA, successful spectral fits can still be achieved for NO$_2$, but the cloud retrievals suffer from large errors that cannot be

overcome. Figure 2 shows the rows flagged in the Collection 3 level 1 data over time. By 2017, 38% of the available data was affected by the row anomaly. All rows affected are flagged with a specific 'row anomaly flag' in the QA4ECV OMI NO$_2$ data product, addressing the user needs expressed in section 2.1.

Spurious across-track variability, or stripes, are apparent in current OMI NO$_2$ data products. The stripes appear as discrete jumps in NO$_2$ SCDs from one viewing angle to the other. The origin of the stripes is probably related to small differences in spectral calibration and detector sensitivity from one viewing angle to the other. There is currently no solution via the level 1 data, but application of a destriping correction (e.g. Boersma et al. [2011] reduces the systematic stripes to within acceptable limits. The

magnitude of the NO$_2$ destriping corrections has increased from $0.3 \times 10^{15}$ to $0.5 \times 10^{15}$ molec. cm$^{-2}$ between 2004 and 2016, related to the use of an annual mean (2005) irradiance spectrum as reference in the DOAS spectral fits.





In OMI's visible detector degradation is on the order of 1-1.5% over the mission period (for rows not affected by the RA). For a signal-to-noise ratio of approximately 500, a deterioration of 1.5% leads to only marginal increases in $NO_2$ fitting uncertainties [Zara et al., 2018]. Spectral stability, important for the accuracy of DOAS retrievals, has also been very good in the visible channel at 0.002 nm. Such

wavelength shifts, if unaccounted for, cause $NO_2$ SCD errors of less than 1%. For more details, please see section 2.2 of QA4ECV Deliverable 4.2 [Müller et al., 2016] and Schenkeveld et al. [2017].

**GOME-2(A)**

GOME-2 on EUMETSAT's MetOp-A satellite is an improved version of the GOME instrument. Level-

1 data, version 6.0, is available from January 2007 onwards. A key concern is the accuracy of the long-term record of GOME-2(A) level-1 data. Like GOME and SCIAMACHY, GOME-2 suffered from degradation of its optical components during its lifetime. The optical parts of GOME-2(A) are thought to be increasingly contaminated by outgassing coating material that was meant to protect the detector electronics [Hassinen et al., 2016]. This contamination resulted in a progressive wavelength-dependent

loss of the instrument throughput. The discontinuity appearing in September 2009 reflects the so-called $2^{nd}$ throughput test, during which the temperature of the GOME-2 instrument was changed in a controlled way to observe whether or not there was a recovery in performance at any point during the heating. Although the test did not recover the degradation already suffered, it did succeed in stabilizing the throughput from September 2009 onwards. The main impact of this degradation is an increase of the

noise due to throughput loss. As a result uncertainties from random error on the $NO_2$ slant columns are expected to increase with time, especially between January 2007 and September 2009. Compared to GOME and SCIAMACHY, degradation of GOME-2(A) started immediately after launch and proceeded faster.



In-flight analysis of the GOME-2(A) instrument slit function using a non-linear fitting of Gaussian lineshapes to the Kurucz solar atlas has revealed significant time variations of the GOME-2 slit function in channel 3 (e.g. Dikty et al., [2011]). Specifically, the nominal width of the slit function (0.50 nm) has decreased over time, probably due to thermal fluctuations of the GOME-2(A) optical bench associated

with seasonal and long-term changes in the solar irradiance [Munro et al., 2015]. In QA4ECV, this issue is addressed by including the GOME-2(A) slit function as a fit parameter in the DOAS spectral fitting procedure. However, it is unlikely that this fully resolves the issue, so that further increases in $NO_2$ SCD uncertainties over time should be anticipated [Zara et al., 2018]. In contrast to GOME and SCIAMACHY, GOME-2(A) solar irradiances are not suffering from spectral structures caused by the

diffuser plate. One minor issue is the sensitivity for polarization structures in the level-1 spectra. In principle, this is corrected for in the level 0-to-1 algorithm [Munro et al., 2015], but some residual small spectral features remain that may interact with atmospheric absorbers in the DOAS fitting.

The instrument specifics, intrinsic quality, and degradation of the 4 instruments' level-1 data have

guided us in selecting the basic settings for spectral fitting of QA4ECV $NO_2$ SCDs. We used these guiding principles:
- Select as much as possible just one or in any case overlapping fitting windows for different instruments. $NO_2$ SCDs are known to be sensitive to the selection of fitting window, as shown in van Geffen et al. [2015], and in the S5P-verification report [S5P/TROPOMI Verification Report,

2015].
- Select a wide fitting window including more $NO_2$ absorption features for an instrument with a relatively low signal-to-noise, i.e. OMI. This is known to reduce the random component of the uncertainty in the $NO_2$ SCDs (e.g. Bucsela et al. [2006]; Boersma et al. [2007]).
- Select the most practical reference spectrum for the DOAS spectral fitting. Ideally these spectra

are daily solar irradiances, as in the case of OMI, but if these are compromised in any way, they may be replaced by an average irradiance spectrum, or by daily Earthshine spectra, as is done for GOME. For the latter, a correction for the amount of $NO_2$ absorption signature in the Earthshine reference spectrum is still required.



## 4 Algorithm design and traceability

An important ambition of the QA4ECV project is to provide full traceability on retrieval algorithms. Usually, a condensed flow diagram for the retrieval algorithm is included in an Algorithm Theoretical Baseline Document (ATBD). The drawback is that ATBDs are often not easily accessible, and that it is not immediately clear which ancillary information has been used in particular algorithm sub-steps. We therefore generated an algorithm 'traceability chain', a web-hosted interactive flow diagram that shows how the QA4ECV $NO_2$ algorithm is put together, which external pieces of information are embedded in the retrieval process, and where details on those pieces of information can be found. The traceability chain has different layers (Figure 3). The main entry for users is the overall algorithm flow chart. Users can click on algorithm process elements, which takes them a level deeper into the algorithm. Figure 3 shows how to interact with the $NO_2$ traceability chain at multiple levels. The chain is provided as a clickable option on the QA4ECV website along with the options 'Data Access' and 'User Forum'. Providing these options at the same 'entrance level' allows users to obtain good understanding of how the algorithm works and where ancillary data is coming from. The 'Traceability Chain' button, leads to the full chain ([1st] layer). Next, as an example, when clicking the 'DOAS + wavelength calibration' step will lead to details on that sub-process ([2nd] layer). The absorption cross sections used in the DOAS step, are available under 'Laboratory Absorption Cross Sections', which contains the references to the cross-section data and papers describing them ([3rd] layer). The references themselves are linked to the digital object identifiers (DOI's) and take users directly to the relevant paper.



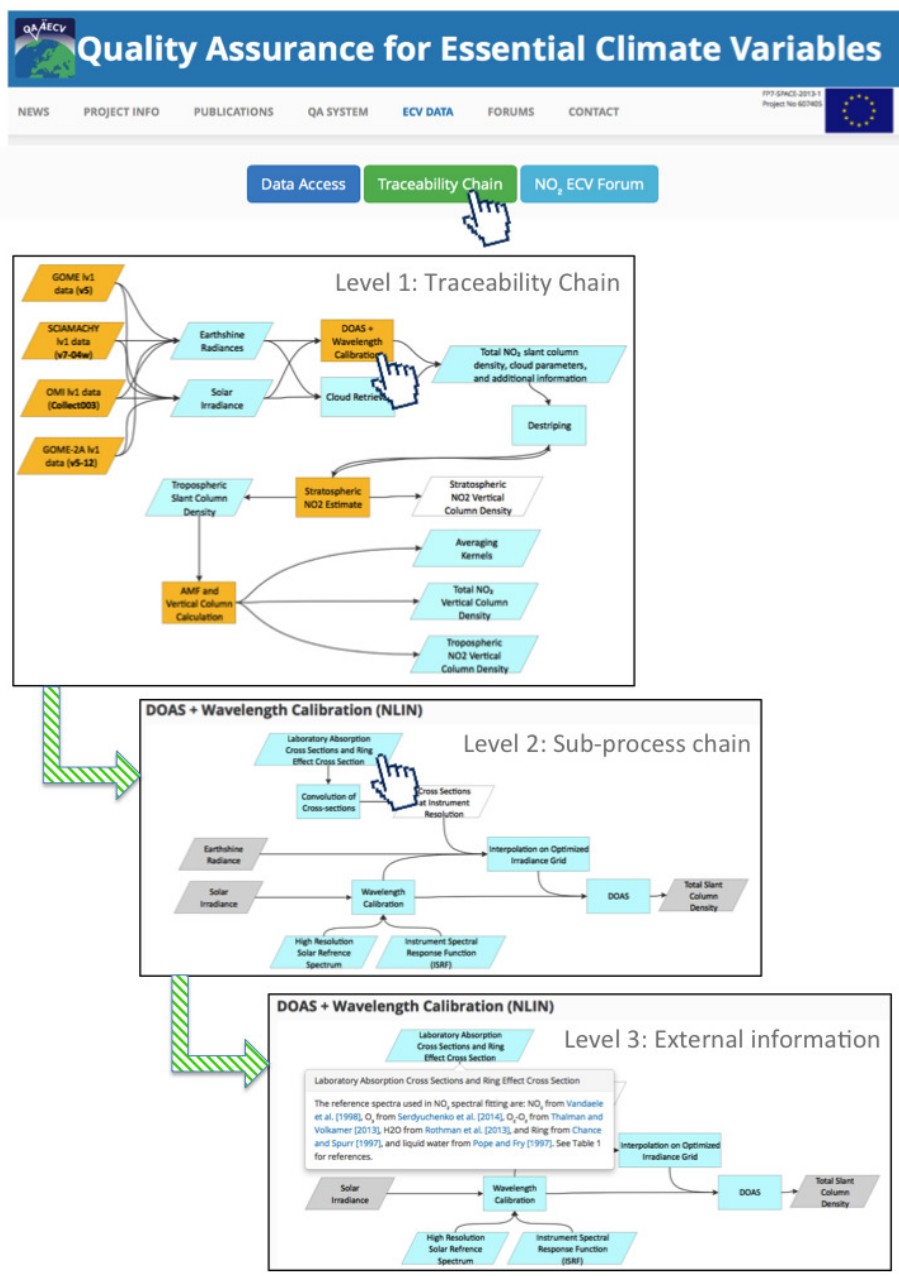

**Figure 3.** Traceability Chain for the QA4ECV $NO_2$ retrieval algorithm. The orange blocks (rectangles) are the building blocks of the retrieval, and in the main chain these are clickable to see more details in deeper layers. The light blue blocks are also clickable and will provide more information on that process in a pop-up window. The parallelograms provide information on algorithm choices and input data sets. The interactive traceability chain is available at: http://www.qa4ecv.eu/ecv/no2-pre.



## 5 Inter-comparison of retrieval sub-steps and algorithm selection

Differences between $NO_2$ retrievals from different retrieval groups can be traced back to different settings and to different a priori parameters used in the individual retrievals. We made a systematic step-by-step analysis of all components of the $NO_2$ retrieval, by documenting and comparing approaches from the consortium institutes, and analysing their contribution to differences and their benefits. These tests, evaluations, and innovations have guided the development of the QA4ECV consortium 'best practice' algorithm for generating a multi-decadal record for $NO_2$, and helped to characterize the uncertainties of each retrieval sub-step.

### 5.1 Evaluation of spectral fitting approaches

$NO_2$ spectral fitting approaches by BIRA-IASB, IUP Bremen, KNMI, and MPI-C were compared in two rounds, with emphasis on OMI and GOME-2:

(1) comparison of $NO_2$ SCDs retrieved by different groups for the same level-1 data with common (as much as possible identical) settings,

(2) as (1), but now with each group using their own, preferred retrieval settings.

The inter-comparison comprised 4 full days: winter and summer early and late in the mission in order to investigate the agreement of retrieval codes with respect to seasonal and instrumental changes. Table 3 shows the details of the spectral fitting retrieval code from the four participating institutes. The retrieval algorithms are based on the same principles, but have been implemented differently, and use different software packages. The KNMI code applies a wavelength shift prior to the DOAS fit, and does not include an intensity offset in the intensity-fitting model. The common settings are listed in the caption of Table 3.

**Table 3.** Overview of OMI SCD retrieval codes from the QA4ECV consortium's institutes. The common settings used for round 1 were: 405-465 nm fitting window, polynomial degree 4, inclusion of $O_3$, $NO_2$, $O_2$-$O_2$, $H_2O$, Ring cross-sections, use of mean solar irradiance as reference spectrum. The cross-sections have been convolved with the OMI slit function for each row separately.



| Institute | Retrieval | Code | Method | Wavelength calibration | Intensity offset | Spike removal | Reference |
|---|---|---|---|---|---|---|---|
| BIRA-IASB | QDOAS | C | Optical depth[3], non-linear least squares regression (Levenberg-Marquard) | Via Fraunhofer atlas, and shift and squeeze | Yes | Yes | Fayt and Van Roozendael [2001] |
| IUP Bremen | NLIN | PASCAL/ DELPHI | Optical depth, non-linear least squares regression (Levenberg-Marquard) | Via Fraunhofer atlas, and shift and squeeze | Yes | Yes | Richter [1997] |
| KNMI | OMNO2A v2 | C | Intensity fit[4], non-linear least squares regression (Levenberg- | Via Fraunhofer atlas, and shift | No | Yes | van Geffen et al. [2015] |

[3]An 'optical depth' fitting model is of the form: $\ln\left(\frac{I(\lambda)}{I_0(\lambda)}\right) = -\sum_i \sigma_i(\lambda)N_{s,i} + \sum_j a_j \lambda^j$ with $I(\lambda)$ the radiance, and $I_0(\lambda)$ the irradiance spectrum, $\sigma_i(\lambda)$ the absorption cross section spectrum of trace gas $i$, $N_{s,i}$ the fitting coefficient, or slant column density of trace gas $i$, and $a_j$ the coefficients of a low order polynomial.

[4]An 'intensity' fitting model is of the form: $I(\lambda) = I_0(\lambda)e^{-\sum_i \sigma_i(\lambda)N_{s,i}+\sum_j a_j \lambda^j}$





| | | | | | | | |
|---|---|---|---|---|---|---|---|
| | | | Marquard) | | | | |
| MPI-C | MPI-C | MATLAB | Optical depth, non-linear least squares regression (Levenberg-Marquard) | Via Fraunhofer atlas, and shift and squeeze | Yes | Yes | Beirle et al. [2013] |

The common settings inter-comparison of OMI $NO_2$ SCDs for all orbits on 2 February 2005, 16 August 2005, 4 February 2013, and 4 August 2013 showed very good agreement between the different algorithms. The correlation between SCDs from each pair of retrieval codes is always >99.8% for all

OMI orbits within the 4 selected days. The correlation is slightly less (but still > 99%) between the KNMI code and the other 3 codes, suggesting that algorithms agree in capturing the full dynamical range of $NO_2$ SCDs. The remaining differences appear over background regions and can be attributed to using a non-linear intensity fitting model instead of a linear optical density fit (resulting in $NO_2$ SCD differences over the oceans up to $1 \times 10^{15}$ molec.cm$^{-2}$, see Figure 4), and to including or excluding an

intensity offset term in the set of fit parameters (differences up to $1 \times 10^{15}$ molec.cm$^{-2}$, reducing contrast between bright and dark scenes). Based on these outcomes, it was recommended to include the intensity offset in the QA4ECV fitting model, even though the exact physical meaning of this term is not completely clear. Including the intensity-offset term appears to account for spectral signatures originating from vibrational Raman scattering in open water, associated incomplete Ring corrections,

and prevents $O_3$ misfits over water and over land. Excluding the intensity offset term results larger $NO_2$ SCD uncertainties, and in (spurious) spatial patterns in the $O_3$ SCDs that resemble the spatial patterns in TOA reflectance. For more details, see QA4ECV Deliverable 4.2, section 2.3 [Müller et al., 2016].





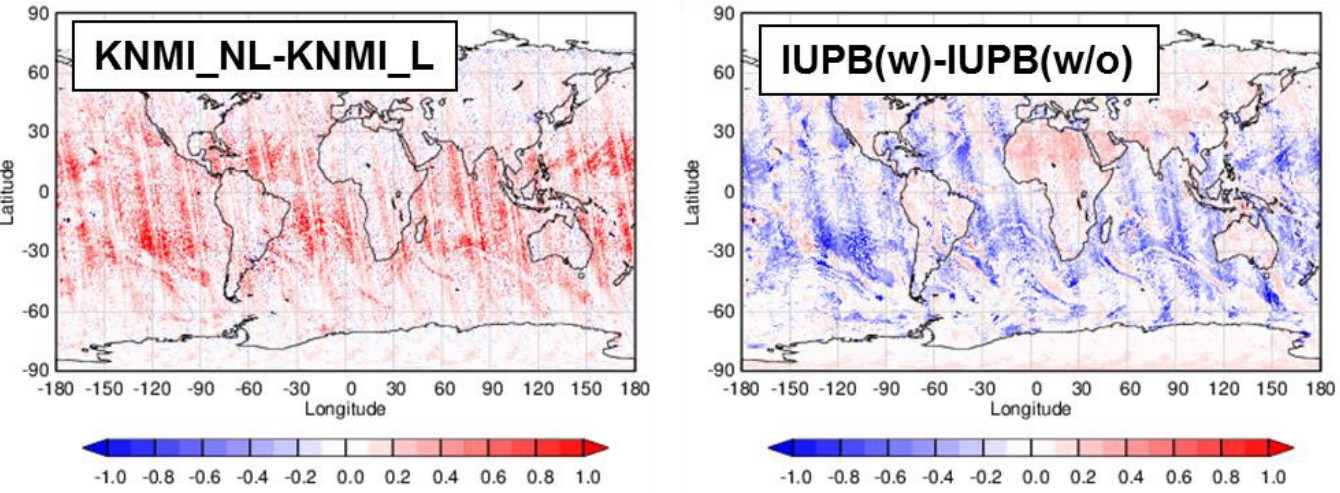

**Figure 4.** OMI NO$_2$ slant column differences between KNMI intensity (KNMI_NL) and optical density fit (KNMI_L) and IUPB fit including (IUPB(w)) and excluding the intensity offset (IUPB(w/o))(right). Data from 2 February 2005.

In round 2, each institute applied preferred settings to retrieve OMI NO$_2$ SCDs for the same set of days. The KNMI settings are identical as in round 1 (Table 3). Relative to the common settings, IUP Bremen used the 425-497 nm fitting window, and included a signature for sand absorption (see Richter et al. [2011]) in the fitting model, BIRA-IASB applied a 425-460 nm window, and included both sand and CHO-CHO signatures, and MPI applied a 431-460 nm window, and excluded liquid water absorption

from the fit. The inter-comparison preferred settings SCDs again showed very good agreement between the algorithms. The correlation between the different pairs is >98%, and the average differences between the different sets are <1×10$^{15}$ molecules cm$^{-2}$. The largest offset (+0.9×10$^{15}$ molecules cm$^{-2}$) appears between KNMI and IUP Bremen (Figure 5(a)). The higher KNMI SCDs are explained by the intensity fit used by KNMI (left panel Figure 4) and by the relatively large difference in centre

wavelengths of the fitting window between these algorithms (435 nm for KNMI, 461 nm for IUP Bremen). Between 405-435 nm, the O$_3$ optical thickness is smaller, and photon paths through the stratosphere are slightly longer than in the 435-500 nm spectral region, located in the flanks of the Chappuis band. DAK simulations indeed show 1.5% higher air mass factors at 405 nm than at 500 nm (Figure 5(b)). For the majority of SCDs retrieved over unpolluted regions, the use of an intensity fit,





together with the 'bluer' fitting window explains the differences between KNMI and IUP Bremen retrievals. It was not possible to point out a clear 'winner' among the different fitting approaches, but including an intensity offset, and liquid water absorption in the fit model, reduced fitting residuals and improved $NO_2$ and $O_3$ fit results. $NO_2$ SCDs are most sensitive to the fitting approach, i.e. intensity fit

or optical density fit.

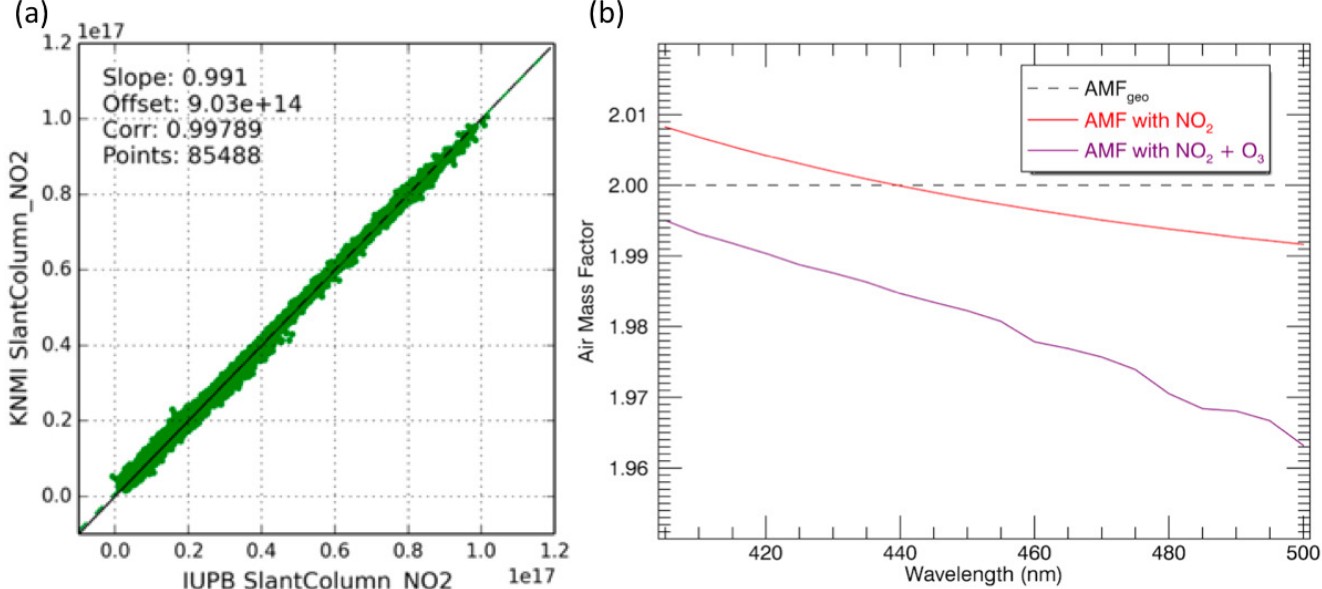

**Figure 5.** (a) Correlation plots of IUP Bremen (425-497 nm fitting window) and KNMI (405-465 nm) $NO_2$ slant columns retrieved using preferred fit settings for OMI orbit OMIL2_2005m0202t0339, including only pixels with SZA < 88° and intensity < $1.1 \times 10^{14}$. (b) Wavelength dependency of the total air mass factor for a scenario with SZA=VZA=0° (geometrical AMF = 2), as calculated with DAK for a

mid-latitude standard atmospheric profile with a total $NO_2$ column of $5.9 \times 10^{15}$ molecules cm$^{-2}$ (mostly situated in the stratosphere) (red curve), and for the same mid-latitude standard atmospheric profile but now with absorption by both $NO_2$ and $O_3$ (total column of 322 DU, purple curve).

The comparisons of the fitting approaches led to a number of clear recommendations for spectral fitting

of $NO_2$ for the QA4ECV record. A complete list can be found in QA4ECV Deliverable 4.2 [Müller et al., 2016]. We highlight the most important ones here:

- Include an intensity-offset correction.



- Given the sensitivity to selecting intensity fit or optical density fit (systematic bias up to $1\times10^{15}$ molecules cm$^{-2}$), it is recommended to use one and the same fit model for all sensors.
- For the 405-465 nm fitting window, the absorption spectrum of liquid water should be included (not necessary for the smaller windows)

Together with the recommendations driven by level-1 data quality considerations shown in Table 3, this has led to the following definition of spectral fitting of NO$_2$ and data processing from GOME, SCIAMACHY, OMI, and GOME-2.

10 **Table 4.** Recommended settings for QA4ECV NO$_2$ spectral fitting for the retrieval of NO$_2$ slant columns from GOME, SCIAMACHY, OMI, and GOME-2(A) to generate a multi-decadal data record for the period 1995-2017.

|  | **OMI** | **GOME-2(A)** | **SCIAMACHY** | **GOME** |
|---|---|---|---|---|
| DOAS processor | QDOAS version Globalcalib6 | NLIN 7.25 (2007-2011) QDOAS (2012-2016) | NLIN 7.35-7.37 | NLIN 7.55 |
| Fitting window | 405-465 nm | 405-465 nm | 425-465 nm | 425-465 nm |
| Fitting method | Optical density | Optical density | Optical density | Optical density |
| Selection reference spectrum | Annual mean (2005) solar reference | Daily solar reference | Daily solar reference (from azimuthal scan mirror diffuser) | Daily Pacific radiance[5] plus offset of 1.476 $10^{15}$ molec. cm$^{-2}$) |
| Polynomial | 4th order | 4th order | 4th order | 4th order |

---

[5]The Pacific Ocean reference sector has been defined as the area enclosed by 160-260°E, 10°S-10°N. The offset has been determined from a comparison with coincident SCIAMACHY SCDs (2002-2003)




| Fitting parameters | $O_3$, $NO_2$, $O_2$-$O_2$, $H_2O$, Ring, liquid water | $O_3$, $NO_2$, $O_2$-$O_2$, $H_2O$, Ring, liquid water | $O_3$, $NO_2$, $O_2$-$O_2$, $H_2O$, Ring, liquid water | $O_3$, $NO_2$, $O_2$-$O_2$, $H_2O$, Ring, liquid water |
|---|---|---|---|---|
| Undersampling correction | No | No | No | Yes |
| Eta correction | No | No | No | Yes |

## 5.2 Evaluation of stratosphere-troposphere separation

We compared stratospheric correction approaches by IUP Bremen, KNMI, and MPI-C to establish best practices for this algorithm step. The stratospheric correction approach from IUP Bremen is based on scaling model-simulated (B3dCTM model) stratospheric vertical columns to match satellite observations over the remote Pacific [Hilboll et al., 2013]. In the KNMI-approach, $NO_2$ SCDs are assimilated in the TM4 model, so that model simulations of stratospheric $NO_2$ columns agree well with the retrieved slant columns over regions away from strong tropospheric pollution [Dirksen et al., 2011]. MPI-C uses a modified reference sector approach called STREAM [Beirle et al., 2016]. This approach estimates the stratospheric vertical columns from retrievals over regions where tropospheric $NO_2$ is assumed to be negligible, and over regions with high clouds, where the tropospheric column is shielded. The derived stratospheric field is then smoothed and interpolated globally, based on the assumption that the spatial pattern of stratospheric $NO_2$ does not feature strong gradients.

The inter-comparison of stratospheric correction approaches focused on 2 individual days (1 January 2005 and 19 July 2005) and 2 monthly means (January and July 2005). This comparison should be regarded as a 'preferred settings' round, where SCD inputs were identical, but the stratospheric AMFs, and methods to estimate the stratospheric $NO_2$ columns varied between the groups. We evaluated the "success" of the stratospheric corrections via checks on the smoothness of stratospheric patterns, and on the plausibility of the tropospheric residues (defined as $N_v$-$N_{v,strat}$) over remote regions where values are expected to be low and not strongly negative. The comparisons (Section 2.4 of QA4ECV Deliverable



4.2 [Müller et al., 2016]) indicated that the different schemes showed similar stratospheric $NO_2$ columns and tropospheric residues, and each of the approaches would be appropriate for use in the QA4ECV $NO_2$ algorithm. The quantitative differences between the stratospheric $NO_2$ columns were generally smaller than $0.5 \times 10^{15}$ molecules $cm^{-2}$, a number that can be regarded as an upper limit for the 'structural

uncertainty' in the stratospheric estimate, but the patterns also revealed that IUP Bremen and KNMI stratospheric $NO_2$ columns were biased high at high solar zenith angles in the winter hemisphere. In Lorente et al. [2017], we attributed this bias to the SCIATRAN and DAK radiative transfer models not fully accounting for the sphericity of the atmosphere in describing photon transport after backscattering. The model used by MPI-C to compute stratospheric AMFs (McArtim) does account for the sphericity of

the atmosphere for both incoming and backscattered light, resulting in lower stratospheric AMFs especially for extreme solar zenith angles.

The KNMI data assimilation was selected as the default approach for estimating the stratospheric $NO_2$ column in the QA4ECV algorithm. This ensures consistent knowledge of the state of the atmosphere ($NO_2$ and temperature profiles, stratospheric dynamics) derived from the same model that predicts the a priori tropospheric $NO_2$ profile shape required by the tropospheric AMF calculation, i.e. TM5-MP

[Williams et al., 2017]. Moreover, the data assimilation approach incorporates a correction for sphericity via McArtim, as described in Lorente et al. [2017]. Retrieval results point out that the stratospheric AMFs, together with improvements in the data assimilation scheme, lead to much fewer negative  tropospheric columns for retrievals at extreme viewing geometries, also at mid-latitudes. As a

second option, the consortium selected MPI-C STREAM stratospheric $NO_2$ estimates to be included in the QA4ECV data product. This allows QA4ECV data users to switch approaches, which may be beneficial for particular applications. Furthermore, the differences between the two methods are useful as a measure of structural uncertainty in the stratospheric correction, beyond the typical uncertainties of $0.2 \times 10^{15}$ molecules $cm^{-2}$ derived from the Observation – Forecast statistics of the assimilation scheme

[Dirksen et al., 2011]. Regions of enhanced structural uncertainty are relevant especially over areas with small tropospheric $NO_2$ enhancements, such as from outflow of continental pollution over oceans, shipping lanes, and over areas with soil $NO_x$ emissions.



As an example, Figure 6 shows OMI stratospheric $NO_2$ estimates from both the data assimilation and STREAM approach for the QA4ECV v1.1 product on 2 February 2005. The upper panel illustrates that the latitudinal gradients in $NO_2$ agree reasoably well between the data assimilation and STREAM agree reasonably well. It is evident that the data assimilation approach captures more variability along a zonal band, resulting on this day in lower stratospheric $NO_2$ over North America and Europe, and higher over northeastern Asia than in the STREAM method. The differences are up to $1\times10^{15}$ molec. cm$^{-2}$, such that they have a substantial impact on the tropospheric column retrievals.

**Figure 6.** Stratospheric $NO_2$ columns from OMI on 2 February 2005, estimated with the data



assimilation method (upper left), and with the STREAM method (upper right). The lower panel shows the differences between the stratospheric $NO_2$ estimates.

## 5.3 Evaluation of air mass factor calculations

We performed a comparison of approaches to calculate AMFs for $NO_2$ and mapped the uncertainties associated with these approaches. Much of this comparison has been reported in Lorente et al. [2017] and in Section 2.5 of QA4ECV Deliverable 4.2 [Müller et al., 2016], so we give only a brief summary here. First, we compared radiative transfer models from the consortium (LIDORT, SCIATRAN, DAK, McArtim) for their top-of-atmosphere reflectances and their capacity to compute vertically resolved, or 'box' AMFs. The agreement between reflectances from the 4 models' at 440 nm (and also at 340 nm) was excellent. Mean relative differences between models were generally small (<1%), with the exception of high solar zenith angles (>80°), where systematic differences with the McArtim model amount to up to 10%. McArtim is the only model that simulates radiative transfer in full sphericity for the direct and the diffuse light [Deutschmann et al., 2011]. Other differences such as different layering schemes, polarization description, refractive index, and Rayleigh scattering cross section spectrum, only lead to small differences (<1%) between the models.

To establish the QA4ECV $NO_2$ algorithm settings, we selected the appropriate wavelength for calculating the $NO_2$ box AMFs. We investigated the wavelength dependency of the $NO_2$ AMFs for retrieval scenarios with substantial tropospheric pollution ($N_{v,trop} = 16 \times 10^{15}$ molec.cm$^{-2}$), and considered that the AMF calculated at a single wavelength should be representative for the fit window average AMF. Tropospheric $NO_2$ AMFs were calculated between 400-500 nm with 1 nm steps. Figure 7 shows a distinct increase of AMF with wavelength. This increase reflects the increasing transparency of the lower troposphere towards the 'green' part of the spectrum where Rayleigh scattering is weakening. In general, tropospheric AMFs increase by 0.2-0.3% per nm redshift. The purple, blue, and light blue lines show the AMF averaged over all spectral points in 3 relevant fitting windows used within QA4ECV and by individual groups.



**Figure 7.** $NO_2$ tropospheric air mass factor (black)[6] as a function of wavelength computed with DAK for a polluted boundary layer for a specific viewing geometry ($\theta$=60°, $\theta_o$=45.6°). Horizontal lines show averaged multi-wavelength AMF for different fitting windows (purple, 425-450 nm, blue 405-465 nm and light blue 425-497 nm). The grey line shows $NO_2$ absorption cross-section from Vandaele et al. (1998) at 220 K. A mid-latitude standard atmosphere was used including $O_3$. The AMF was computed for a polluted boundary layer with $16\times10^{15}$ molec/cm², without aerosols, a boundary layer height of 1 km and surface albedo 0.05.

---

[6]The non-smooth behaviour of the red line is because the spectral resolution of the AMF is not sufficient to resolve the $NO_2$ cross section used in the calculation. If a constant cross section value is used in the RTM for calculating TOA reflectance the increasing AMF with wavelength would be spectrally smooth.



We saw in Figure 5 that stratospheric $NO_2$ AMFs weakly decrease with wavelength (-0.01%/nm redshift). Figure 7 shows that tropospheric $NO_2$ AMFs increase with wavelength (+0.2-0.3%/nm). This difference can be understood from Rayleigh scattering, occurring mostly in the lowest kilometres of the

troposphere. The bulk scattering increasingly screens $NO_2$ in the boundary layer towards the UV, so that tropospheric AMFs are smallest for shorter wavelengths. For the fitting windows considered for QA4ECV $NO_2$ retrievals (425-465 nm and 405-465 nm), we recommend calculating the $NO_2$ AMF at 437.5 nm for all sensors. The blue and purple lines in Fig. 8 indicate that 437.5 nm is a representative wavelength to calculate the $NO_2$ AMF. 437.5 nm is reasonably near to the centre wavelength of both

windows (435 and 445 nm respectively) and the 437.5 nm AMF is within 2% of the window-average AMF for both windows. Uncertainties related to the exact choice of AMF wavelength calculation are much smaller than other AMF uncertainties, such as clouds, albedo, trace gas and aerosol profiles, as discussed below and in Lorente et al. [2017].

We compared altitude resolved AMFs and tropospheric AMFs calculated with the 4 different radiative transfer models. We found that the agreement is very good (within 3% and 6% respectively), if identical ancillary data (surface albedo, terrain height, cloud parameters and trace gas profile) and cloud and aerosol corrections are being used. This shows that the choice of RTM for calculation of the tropospheric AMF introduces a modest uncertainty of no more than 6%, which is intrinsic to the

calculation method, and cannot be avoided.

To assess the full impact of preferred settings and methods for AMF calculations, we organized a round robin comparison. Six groups joined this round robin, each using their preferred setting to calculate the tropospheric AMFs. Besides the QA4ECV-partners KNMI/WUR, BIRA-IASB, and IUP Bremen, also

NASA GSFC, Leicester University, and Peking University participated. The 6 groups used widely different calculation methods (RTMs, temperature, cloud and aerosol corrections) and preferred ancillary data on albedo, terrain height, $NO_2$ profile, etc. [Müller et al., 2016]. The ensemble mean AMF served as a reference against which to compare the AMFs by the individual groups. The round robin





exercise focused on China because it provides challenging retrieval conditions and Peking University only calculates AMFs over that region. The overall spatial pattern of AMF values was well-reproduced by all groups. AMFs generally agree to within 10% over unpolluted areas, but show differences of up to 40% with respect to the ensemble mean over polluted regions in eastern China and Korea. These

differences can be traced back to differences in the preferred surface albedo, clouds, and a priori $NO_2$ profiles used in the AMF calculation. It is not possible to identify the single most important forward model dependency for the AMF calculation. The analysis in QA4ECV Deliverable 4.2 [Müller et al., 2016] and in Lorente et al. [2017] suggests that accurate knowledge on surface albedo, clouds, and a priori $NO_2$ profiles are of similar importance, and their interplay, in combination with the choices for

cloud and aerosol correction methods is driving the structural uncertainty in the $NO_2$ AMFs.

Based on the results from the comparisons discussed above, the following recommendations for calculating QA4ECV $NO_2$ AMFs were made:

- calculate the $NO_2$ AMFs at 437.5 nm for all instruments

- apply the independent pixel approximation for cloud correction, but also include clear-sky AMFs in the product

- use cloud information (cloud fraction, cloud pressure) from FRESCO+ for GOME, SCIAMACHY, GOME-2A [Wang et al., 2008] and OMCLDO2 for OMI [Veefkind et al., 2016]. These have been derived using the same physical principles as in the AMF calculation.

- apply implicit aerosol correction (via the cloud correction). This correction is effective in most retrieval scenarios with moderate aerosol pollution. When accurate, observation-based aerosol information becomes available from e.g. ECMWF CAMS or NASA GMAO, explicit aerosol corrections will be considered.

- use surface albedo climatologies (as close as possible to the 437.5 nm AMF wavelength)

consistent with the ones used in the cloud retrievals. For GOME, SCIAMACHY, and GOME-2A, this is the albedo climatology from Tilstra et al. [2017], and for OMI the updated 5-year climatology [Kleipool et al., 2008].

- use the DEM_3KM pixel-average terrain height.





- use spatially interpolated (to pixel centre) $NO_2$ profiles simulated by TM5-MP at 1°×1°. TM5-MP is the model used for the data assimilation of $NO_2$ SCDs to estimate the stratospheric contribution (section 5.2).

## 6 QA4ECV $NO_2$ uncertainty estimates

### 6.1 Theoretical algorithm uncertainty

The QA4ECV $NO_2$ product contains an algorithm uncertainty estimate associated with each individual pixel. This estimate is calculated theoretically via uncertainty propagation based on the principal retrieval equation (Eq. (1)):

$$\sigma = \sqrt{\left(\frac{\sigma_{N_S}}{M_{tr}}\right)^2 + \left(\frac{\sigma_{N_{S,strat}}}{M_{tr}}\right)^2 + \left(\frac{(N_S - N_{S,strat})\sigma_{M_{tr}}}{M_{tr}^2}\right)^2} \tag{3}$$

The uncertainty propagation accounts for spectral fitting uncertainties ($\sigma_{N_S}$), and contributions from uncertainties in a priori and ancillary data required for calculating the stratospheric $NO_2$ background ($\sigma_{N_{S,strat}}$) and the AMF ($\sigma_{M_{tr}}$). The uncertainty in the tropospheric AMF, or AMF covariance is written as:

$$\sigma_{M_{tr}}^2 = \left(\frac{\partial M}{\partial A_s}\sigma_{A_s}\right)^2 + \left(\frac{\partial M}{\partial f_{cl}}\sigma_{f_{cl}}\right)^2 + \left(\frac{\partial M}{\partial p_{cl}}\sigma_{p_{cl}}\right)^2 + (0.1 M_{tr})^2 + 2\left(\frac{\partial M}{\partial A_s}\frac{\partial M}{\partial f_{cl}}\langle\epsilon_{f_{cl}}\epsilon_{A_s}\rangle\right) \tag{4}$$

where $\frac{\partial M}{\partial A_s}$ represents the local sensitivity of the the air mass factor to surface albedo $A_s$, and $\sigma_{A_s}$ the best estimate of the uncertainty in the surface albedo, and so on. The fourth term on the right hand side represents the contribution from uncertainty in the a priori profile shapes, and is tentatively approximated as 10% of the tropospheric AMF. This term is absent when using the averaging kernel in satellite data applications [Eskes and Boersma, 2003], which removes the dependence on the a-priori





profile. The last term represents the contribution from error correlation between cloud fractions and surface albedo $\langle \epsilon_{f_{cl}} \epsilon_{A_s} \rangle$; surface albedo influences AMF directly, and indirectly because cloud fractions are sensitive to surface reflectance (see Eq. (20) and Eq. (A2) in Boersma et al. [2004] and Lorente et al. [2018] for more detail). As $\langle \epsilon_{f_{cl}} \epsilon_{A_s} \rangle$ and $\frac{\partial M}{\partial f_{cl}}$ are negative, and $\frac{\partial M}{\partial A_s}$ is positive, this last term gives a

positive contribution to $\sigma_{M_{tr}}^2$.

The uncertainty $\sigma$ should be interpreted as the best guess of the retrieval uncertainty for one specific measurement. This uncertainty contains random and systematic error components, and the different systematic error components (due to errors in profile shape, surface albedo, etc.) each have their own

spatial and temporal scale. Therefore, when averaging over multiple pixels (spatially) or over time, part of the error will cancel out or be smoothed, but (an unknown) part of the systematic error will remain even after averaging, see Boersma et al. [2016].

We recommend using Eq. (5) below to estimate the uncertainty $\sigma_o$ for spatially or temporally averaged

data. This method takes the area-weighted (statistical) retrieval uncertainty $\sigma$, and then accounts for a partial correlation in the errors between pixels as in Eskes et al. [2003]:

$$\sigma_0 = \sigma \sqrt{\frac{1-c}{n} + c} \qquad (5)$$

with $c$ the error correlation between the $n$ retrievals. In Boersma et al. [2016], $c=0.15$ is proposed based on the consideration that errors in surface albedo, clouds, a priori $NO_2$ profile, and aerosols (or lack of description thereof) are typically correlated at the spatiotemporal scales of moderate resolution (global) models, i.e. down to $0.5° \times 0.5°$ and over one month (for example the surface albedo is from a monthly climatology). Eq. (5) with $c=0.15$ implies that the spatially or temporally averaged uncertainty cannot

reduce to below 39% of the level of typical single-pixel uncertainties ($\sigma$), even when many observations are available.



## 6.2 Algorithm uncertainties and quality flags

Table 5 gives an overview of the most important uncertainties and the quality flags of QA4ECV NO$_2$ provided in the data product. Note that the uncertainty estimates and quality flags provide clearly different information to the user. The uncertainty characterizes the dispersion of the NO$_2$ column, given
5  the value of the measured column, and our best understanding of the retrieval process. Quality flags indicate whether the retrieved value and the uncertainty estimate have been obtained under conditions where they are expected to be valid.

**Table 5.** Overview of the main uncertainty estimates and quality flags included in the QA4ECV NO$_2$
10  ECV precursor product.

| Name | Meaning | Symbol |
|---|---|---|
| Tropospheric NO$_2$ column uncertainty | Per pixel algorithm uncertainty estimate of the tropospheric NO$_2$ column | $\sigma$ |
| Tropospheric NO$_2$ column uncertainty when averaging kernel is applied | Per pixel algorithm uncertainty estimate. Same as above, but contribution from profile uncertainty removed | $\sigma_{AK}$ |
| Stratospheric NO$_2$ column uncertainty | Global estimate of uncertainty in the stratospheric VCD | $\sigma_{N_{strat}}$ |
| Uncertainty of the sum of the tropospheric and stratospheric vertical NO$_2$ columns | Per pixel algorithm uncertainty estimate of the total NO$_2$ column | |
| NO$_2$ SCD uncertainty | Uncertainty estimated from the DOAS spectral fitting of NO$_2$ | $\sigma_{N_S}$ |
| Slant column related uncertainty of the NO$_2$ tropospheric vertical column | First term on the right hand side of Eq. (3) | $\left(\frac{\sigma_{N_S}}{M_{tr}}\right)$ |
| Stratospheric column related | Second term on the right hand side of Eq. | $\left(\frac{\sigma_{N_{s,strat}}}{M_{tr}}\right)$ |





| uncertainty of the NO$_2$ tropospheric vertical column | (3) | |
|---|---|---|
| Total tropospheric AMF related uncertainty of the tropospheric NO$_2$ vertical column | Third term on the right hand side of Eq. (3). | $\left( \dfrac{(N_S - N_{S,strat})\sigma_{M_{tr}}}{M_{tr}^2} \right)$ |
| Surface albedo related uncertainty of the tropospheric vertical NO$_2$ column | Contribution to the uncertainty of uncertainties in the surface albedo in the tropospheric AMF | $\left( \dfrac{\partial M}{\partial A_s} \sigma_{A_s} \right) \dfrac{N_v}{M_{tr}}$ |
| Cloud fraction related uncertainty of the tropospheric vertical NO$_2$ column | Contribution to the uncertainty of uncertainties in the cloud fraction in the tropospheric AMF | $\left( \dfrac{\partial M}{\partial f_{cl}} \sigma_{f_{cl}} \right) \dfrac{N_v}{M_{tr}}$ |
| Cloud pressure related uncertainty of the tropospheric vertical NO$_2$ column | Contribution to the uncertainty of uncertainties in the cloud pressure in the tropospheric AMF | $\left( \dfrac{\partial M}{\partial p_{cl}} \sigma_{p_{cl}} \right) \dfrac{N_v}{M_{tr}}$ |
| TM5 profile related uncertainty of the tropospheric vertical NO$_2$ column | Global estimate of the contribution to the uncertainty of uncertainties in the TM5 NO$_2$ profile in the tropospheric AMF | $0.1 N_v$ |
| processing error flag | Flag indicating whether the processing was successful (0) or failed (-1) | |
| processing quality flags | Flags indicating conditions that affect the quality of the retrieval | |

## 6.3 Evaluating the sub-step uncertainty estimates

An innovative aspect of the QA4ECV project is the evaluation of the uncertainty estimates of retrieval sub-steps against independent estimates of the same metric and structural uncertainties.



### 6.3.1 Evaluation of NO$_2$ SCD uncertainties

We compared the DOAS uncertainty estimates ($\sigma_{N_S}$) from the spectral fitting algorithm against independent estimates obtained from the spatial variability of an ensemble of DOAS SCDs over areas with little geophysical variability using a statistical approach [Boersma et al., 2007]. Our SCD

uncertainty evaluation is described in detail in QA4ECV Deliverable 5.5 [Boersma et al., 2017] and in Zara et al. [2018] for OMI and GOME-2A, and we summarize results here. For both instruments, we found that the improved QA4ECV OMI NO$_2$ retrieval shows smaller uncertainties than other OMI algorithms and good agreement between the DOAS and statistical SCD uncertainties. This suggests that the recommendations made in section 5.1 and in QA4ECV D4.2 [Müller et al., 2016] have improved the

spectral fitting of NO$_2$ such, that the typical mission-average SCD uncertainties for both instruments amount to 0.7-0.8×10$^{15}$ (was ~1.0×10$^{15}$) molec. cm$^{-2}$. For OMI, this uncertainty is dominated by random contributions from propagation of measurement noise, but we also noticed a 30% systematic contribution from stripe effects. For OMI, the trend in SCD uncertainties was small (<2%/yr) in line with the known radiometric stability of the instrument [Schenkeveld et al., 2017], but for GOME-2A,

the NO$_2$ SCD uncertainties increased by 8%/yr until September 2009, and after heating the instrument by <3%/yr over 2009-2015. The structural (systematic) uncertainty, estimated from the differences between NO$_2$ SCDs calculated with different but equally plausible fitting methods (with or without intensity offset correction, see section 5.1) is larger than but of similar magnitude as the theoretical and statistical estimates. Table 6 gives an overview of the various estimates of uncertainty for the NO$_2$

SCDs.

### 6.3.2 Evaluation of uncertainties in the stratospheric correction

The uncertainty of the stratospheric NO$_2$ vertical column in QA4ECV NO$_2$ product is based on a global statistical analysis of results from the data assimilation procedure, and documented as 0.2×10$^{15}$

molecules cm$^{-2}$ [Dirksen et al., 2011]. The assimilation predicts stratospheric NO$_2$ columns from an observation-constrained (analyzed) start field and TM5-modeled transport and chemistry. The average discrepancies between the 24-hour forecast and actual satellite-observed NO$_2$ slant column fields over




pristine areas are regarded as a measure for the uncertainty in the stratospheric $NO_2$ field. In QA4ECV Deliverable 5.5 [Boersma et al., 2017], we verified that the observation minus forecast (O-F) assimilation statistics over the Pacific are indeed consistent with an uncertainty estimate of $0.2 \times 10^{15}$ molecules $cm^{-2}$ for the stratospheric column.

To further evaluate the estimate of the stratospheric column uncertainty, we compare the QA4ECV data assimilation and STREAM OMI stratospheric $NO_2$ column estimates for 2 February 2005. There are considerable methodological differences between the data assimilation and STREAM techniques. Yet the data assimilation and STREAM stratospheric $NO_2$ distributions agree to reasonable extent, with data

assimilation stratospheric columns generally smaller and their spatial features sharper than in STREAM. The TM5-MP assimilation approach distinguishes stratospheric $NO_2$ from free-tropospheric background contributions, while STREAM does not do this. This may be a main reason for the structurally lower values in the assimilation. This is illustrated in Figure 8, which shows the meridional variability in the stratospheric $NO_2$ column from data assimilation and from STREAM along 40°N on 2 February 2005.

Between 75°-125°W over the United States and between 0-40° E (Europe), the data assimilation stratospheric columns values are $0.2\text{-}0.5 \times 10^{15}$ molecules $cm^{-2}$ lower than the STREAM values. Over eastern Asia (100-140° E), data assimilation and STREAM agree to within $\pm 0.3 \times 10^{15}$ molecules $cm^{-2}$. These differences reflect the structural uncertainty in stratospheric (vertical) $NO_2$ columns, arising when different retrieval methodologies are applied to the same satellite observations, and both uncertainty

estimates are included in Table 6.







**Figure 8.** Meridional average QA4ECV OMI $NO_2$ column averaged over 39-41°N on 2 February 2005. No cloud radiance, albedo, or AMF filtering has been applied. Data assimilation and STREAM stratospheric columns are indicated in the black and green lines, the total slant columns divided by the geometric AMF is light blue. Both data assimilation and STREAM stratospheric column estimates are included in the QA4ECV $NO_2$ product.

## 6.3.3 Evaluation and breakdown of uncertainties in the air mass factors

The uncertainty in the tropospheric AMF is calculated via the uncertainty propagation from Eq. (4). The contribution of each parameter to the overall AMF uncertainty depends on the specific observation



conditions for each pixel. The air mass factor sensitivities (e.g. $\frac{\partial M}{\partial A_s}$) describe the sensitivity of the AMF to changes in the local parameter value, evaluated around the specific value for the parameter at the pixel. The uncertainties in the cloud parameters ($\sigma_{f_{cl}}, \sigma_{p_{cl}}$), surface albedo ($\sigma_{A_s}$), and the a priori profile shape have been estimated from the literature or derived from comparisons with independent data. For

QA4ECV OMI NO$_2$, we use an uncertainty in the surface albedo of 0.015, based on various comparisons of albedo databases (e.g. Boersma et al., [2011]), uncertainties of 0.025 and 50 hPa in the OMI O$_2$-O$_2$ cloud fraction and cloud pressure estimates, respectively, based on recent improvements in the cloud algorithm [Veefkind et al., 2016], and a 10% contribution from NO$_2$ profile uncertainty. The latter is based on comparing AMFs calculated with simulated a priori profiles to AMFs calculated with

measured NO$_2$ profiles from aircraft and lidar (e.g. Hains et al. [2010] and references therein).

Apart from the overall AMF uncertainty estimate, the QA4ECV NO$_2$ ECV precursor data product also provides the individual contributions by the cloud parameters, surface albedo, and a priori profile shapes. Figure 9 presents the relative monthly average tropospheric AMF uncertainties and their

individual contributions from surface albedo, cloud and profile uncertainties (not shown because they have been set at the 10% level) for OMI throughout 2005 over Europe, the United States, China and Johannesburg, South Africa, regions polluted with NO$_2$. The largest contribution to AMF uncertainty is from surface albedo-cloud cross-term contribution (10-20%), with surface albedo contributions contributing substantially (±10%). In winter the uncertainty in cloud pressure is a substantial contributor

in Europe and China. The strong surface albedo-cloud fraction cross-term ($2\left(\frac{\partial M}{\partial A_s}\frac{\partial M}{\partial f_{cl}}\langle\epsilon_{f_{cl}}\epsilon_{A_s}\rangle\right)$) can be understood from the strong sensitivity of the cloud fraction to the surface albedo, especially when cloud fractions are small (see Appendix in Boersma et al. [2004]). The overall tropospheric AMF uncertainties are estimated to be 20-25%, comparable to earlier estimates for GOME tropospheric NO$_2$ presented in Boersma et al. [2004].



**Figure 9.** Average (single-pixel) QA4ECV OMI tropospheric AMF uncertainty (black line) estimated for Europe (40-55°N, 10°W-15°E), United States (35-45°N, 100°W-75°W), China (35-45°N, 110-140°E), and Johannesburg (24-28°S, 26-30°E) in 2005. The pink line indicates the contribution to AMF uncertainty from surface albedo-cloud fraction error correlations, the red line indicates the AMF uncertainties due to surface albedo uncertainty. The contribution from a priori profile uncertainty is assumed to be constant at 10% of the AMF uncertainty (not plotted).

We quantified the structural uncertainty in tropospheric AMFs by comparing an ensemble of different AMF calculation methods and parameter assumptions over eastern China, a region with high amounts and a complex mixture of aerosols, clouds, and $NO_2$ pollution [Lorente et al., 2017]. Retrieval groups

5    used their preference for ancillary data and preferred cloud and aerosol corrections. The outcome of the



comparisons suggested systematic AMF differences of up to 15% in Summer and 40% in Winter between the groups. We consider these structural uncertainty estimates to be conservative, as they have been calculated for the particularly challenging retrieval regime of eastern China in 2005. Including the structural uncertainties in the overall budget, as done for the QA4ECV HCHO ECV precursor product

[De Smedt et al., 2017], would bring tropospheric AMF uncertainties to ±30% in Summer and ±50% in Winter.

**Table 6.** Comparison of uncertainty estimates for the main QA4ECV OMI $NO_2$ retrieval steps. The SCD and stratospheric SCD uncertainties are representative for all possible retrieval scenarios. AMF

uncertainties are representative for situations with high $NO_2$.

| | Algorithm uncertainty (molec. cm$^{-2}$) | Independent uncertainty estimate (molec. cm$^{-2}$) | Structural uncertainty (molec. cm$^{-2}$) |
|---|---|---|---|
| SCD ($\sigma_{N_S}$) | $0.8 \times 10^{15}$ [a] | $0.7 \times 10^{15}$ [a] | $1.0 \times 10^{15}$ [b] |
| Stratospheric SCD ($\sigma_{N_{s,strat}}$) | $0.2 \times 10^{15} \cdot M_{strat}$ [c] | | $(0.2\text{-}0.5) \times 10^{15} \cdot M_{strat}$ [d] |
| Tropospheric AMF ($\sigma_{M_{tr}}$) | 20% (Summer) 25% (Winter) | | 15% (Summer)[e] 40% (Winter)[e] |

[a]Zara et al. [2018]

[b]Section 5.1 of this work

[c]Dirksen et al. [2010] and analysis of data assimilation observation minus forecast differences QA4ECV Deliverable 5.5 [Boersma et al., 2017].

[d]Figure 8 of this work.

[e]Lorente et al. [2017]

## 6.4 Overall uncertainties in tropospheric $NO_2$ columns

## 6.4.1 Uncertainties in single-pixel tropospheric $NO_2$ columns





Here we present estimates of typical algorithm, single-pixel uncertainties for the QA4ECV NO$_2$ columns in four regions: Europe, United States, and China as showcases for typical polluted regions, and the Pacific Ocean as an example of a remote region, with low, background levels. These uncertainty estimates should be interpreted as representative for typical, single-pixel uncertainties encountered by

5    users interpreting the data. We see from Figure 10 that over the polluted regions in wintertime, the single-pixel retrieval uncertainty is dominated by the uncertainty in the tropospheric AMF. In Summer, contributions from uncertainties in the SCD are largest, but with comparable contributions from uncertainties in the stratospheric correction and the tropospheric AMF. On average a single pixel is 35%-45% uncertain in the polluted regions. Over the background region (Pacific Ocean), we see that

10   the tropospheric NO$_2$ column uncertainty exceeds 100%, and is dominated year-round by the uncertainties in SCD and stratospheric column estimate.





**Figure 10.** Average (single-pixel equivalent) QA4ECV OMI tropospheric $NO_2$ columns (solid line) and associated total uncertainties (dashed black line) for Europe (40-50°N, 10°W-15°E), United States (35-45°N, 100-75°W), eastern China (30-45°N, 110-140°E), and the Pacific Ocean (35-45°N, 160-140°W) in 2005. The dashed coloured lined indicate the contributions to the tropospheric $NO_2$ column uncertainty from SCD (pink, $\left(\frac{\sigma_{N_S}}{M_{tr}}\right)$), stratospheric correction (light blue, $\left(\frac{\sigma_{N_{S,strat}}}{M_{tr}}\right)$), and the tropospheric AMF (purple, $\left(\frac{(N_S - N_{S,strat})\sigma_{M_{tr}}}{M_{tr}^2}\right)$).

## 6.4.2 Uncertainties in averaged tropospheric $NO_2$ columns

When averaging tropospheric columns over space, uncertainties may be considerably reduced. For example, over regions such as the Pacific Ocean, where the uncertainty is dominated by random SCD

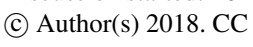



error, the tropospheric column uncertainty will be much reduced when averaging over a month or over a larger region. Over polluted regions, dominated by uncertainties in the tropospheric AMF, averaging will also reduce the tropospheric column uncertainties, but an unknown systematic component will remain. For both retrieval situations, we adopt Eq. (5), to account for possible systematic errors arising

from imperfect knowledge of surface albedo, a priori $NO_2$ profile, clouds, and correlations between these.

In model-column comparisons and in trend analysis studies, it is often important to have knowledge of temporally averaged uncertainties. Because the temporal variability in tropospheric $NO_2$ columns is

typically strong (because of the diurnal cycle, day-to-day variability, weekly cycles, etc.), this implies considerable variability in day-to-day uncertainties. To obtain the uncertainty in a monthly mean tropospheric $NO_2$ column over a certain region, we recommend taking whichever is largest: (a) the temporally averaged values for $\sigma_o$ (Eq. (5)), or (b) the standard deviation of the mean (standard error) of the daily tropospheric $NO_2$ columns. If there is substantial temporal variability (from changes in

photochemistry, transport events), the standard error will be a good representation of the uncertainty in the monthly mean tropospheric $NO_2$ column. Figure 11 shows a comparison of monthly averaged uncertainties $\sigma_o$ and the local standard deviation of the mean $NO_2$ columns for four small regions (0.25° × 0.25°). The figure confirms that the averaged uncertainties provide an optimistic estimate of the uncertainty, at ±10%, in the monthly mean $NO_2$ columns. For the polluted regions, the standard

deviation of the mean is 15-30%, exceeding the average uncertainties. This illustrates that calculating the uncertainty in a monthly mean over a small region such as a city, is more driven by sampling limitations, than by the intrinsic uncertainty of the retrieval.



**Figure 11.** Monthly mean single grid-cell QA4ECV OMI tropospheric $NO_2$ columns (solid line), standard deviation of the mean (standard error, dashed red line), and super-observation uncertainty ($\sigma_0$, dashed black line) in 2005 over Amsterdam (52.375°N, 4.875°E), New York City (40.875°N, 73.875°W), Beijing (39.875°N, 116.375°E), and the Pacific (39.875°N, 149.875°W). Grid cell size 0.25° × 0.25°. Only pixels with cloud radiance fraction < 0.5 were included in the calculation.

## 7 Validation of QA4ECV $NO_2$ columns and uncertainties

As an example of the validation efforts taken within QA4ECV, we here compare the QA4ECV OMI tropospheric $NO_2$ with independent MAX-DOAS column measurements in the polluted city of Tai'an, China. We compare OMI pixels measured within 20 km and 30 minutes of a MAX-DOAS

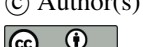



measurement in Tai'an. We validate both the QA4ECV v1.1 and the well-established DOMINO v2 product for reference.

The MAX-DOAS measurements were conducted by Irie et al. [2008] in the Chinese city of Tai'an in
May-June 2006 when pollution levels were substantial. The instrumentation and retrieval technique have been described extensively in Irie et al. [2008; 2009; 2012]. The slant column retrievals have been tested in a semi-blind inter-comparison exercise in Cabauw, The Netherlands, indicating agreement to within 10% with other groups [Roscoe et al., 2010]. Uncertainties in the MAX-DOAS $NO_2$ columns are driven by noise, air mass factor and temperature uncertainties amounting to approximately 15%
uncertainty. The representative horizontal 'footprint' of the MAX-DOAS measurement is on the order of 10 km. It was suggested by Irie et al. [2012] that the spatial distribution of $NO_2$ tropospheric columns around Tai'an during their observation period was rather homogeneous compared to other sites used for their validation comparisons.  More quantitative characterization on this aspect will be discussed below.

We compare OMI $NO_2$ tropospheric columns measured with a pixel center within 20 km of the location of the MAX-DOAS instrument in Tai'an (for some days more than one pixel can be matched up with a MAX-DOAS measurement). This coincidence criterion limits spatial representativeness mismatches between MAX-DOAS and OMI, and is consistent with the spatial dimensions of the MAX-DOAS (±10 km) and OMI (20-30 km) footprints (we excluded pixels from the outer 4 OMI rows). We furthermore
require that the OMI columns were measured within 30 minutes of the coinciding MAX-DOAS measurement, have a pixel footprint area < 700 $km^2$, and that the satellite retrieval was done under mostly clear-sky conditions (cloud radiance fraction <0.5), which is in line with recommendations on the appropriate use of QA4ECV data as documented in the Product Specification Document [QA4ECV Deliverable D4.6]. Earlier studies (e.g. Pinardi et al. [2017]; Drosoglou et al. [2017]) found largest
discrepancies between MAX-DOAS and satellite $NO_2$ columns over strongly polluted regions. Such discrepancies are at least partly due to spatial inhomogeneity in the $NO_2$ field around the station location. To quantify the spatial representativeness of the Tai'an MAX-DOAS site for the OMI pixels included in the comparison, we calculated the campaign-mean spatial tropospheric $NO_2$ column



distribution (Figure 12). We then use the ratio of this campaign-mean column at Tai'an to the campaign-mean column at the location of the individual OMI pixel to project individual OMI $NO_2$ columns ($N_{V,p}$, i.e. what is usually validated) within our criteria to values more representative for the location of the Tai'an ($N_{V,T}$):

$$N_{V,T} = \left(\frac{\overline{N_{V,T}}}{\overline{N_{V,p}}}\right) \cdot N_{V,p} \tag{6}$$

For example, for a pixel observed directly southwest of Tai'an, where pollution levels are somewhat higher than directly over Tai'an, the scaling factor will be smaller than 1. For the coincidence criterion

10 of 20 km used here, the scaling factors stay close to 1 and modifications do not exceed $1\times10^{15}$ molec.cm$^{-2}$ (<20% of the Tai'an column, see Figure S1).





**Figure 12.** Campaign mean (30 May – 30 June 2006) QA4ECV tropospheric $NO_2$ column distribution over eastern China for clear-sky situations (cloud radiance fraction<0.5). The black circle indicates the location of Tai'an, where Chiba University operated the MAX-DOAS instrument. One cell corresponds to $0.1° \times 0.1°$. On average there are 15 satellite pixels per cell used to calculate the campaign mean.

We match each OMI pixel fulfilling the spatio-temporal coincidence criteria with the corrected MAX-DOAS $NO_2$ columns. Discarding pixels with effective cloud pressures >875 hPa (often indicative of aerosol haze), we find 31 QA4ECV OMI pixels matching 13 independent MAX-DOAS measurements collected over 7 different days. Figure 13(a) shows a scatter plot of QA4ECV vs. MAX-DOAS tropospheric $NO_2$ columns for Tai'an. We find a bias (mean difference) of $-0.15\times10^{15}$ molec.cm$^{-2}$ (-2%) and the root mean squared deviation is $1.08\times10^{15}$ molec.cm$^{-2}$ (16%). Not applying the scaling factors



from Eq. (6) leads to a bias of $-0.47 \times 10^{15}$ molec.cm$^{-2}$ (-7%) and a root mean squared deviation of $1.19 \times 10^{15}$ molec.cm$^{-2}$ (18%). Using a reduced major axis regression analysis, we find a relationship between QA4ECV ($y$) and MAX-DOAS NO$_2$ columns ($x$) as $y = -0.86 \times 10^{15}$ molec.cm$^{-2}$ + 1.10$x$ ($R^2$=0.26, $n$=31). Including also pixels with high effective cloud pressures (>875 hPa) leads to a bias of

$-0.48 \times 10^{15}$ molec.cm$^{-2}$ (-6.6%, $n$=37) and a root mean squared deviation of $1.35 \times 10^{15}$ molec.cm$^{-2}$ (20%).

Figure 13(b) shows the scatter plot of DOMINO v2 vs. MAX-DOAS NO$_2$ columns for Tai'an. There are now 45 DOMINO v2 pixels matching with 17 independent MAX-DOAS measurements. This higher number of matches can be explained from the previous version of the OMI O$_2$-O$_2$ cloud product

[Acarreta et al., 2004], used in the DOMINO v2 retrieval, containing effective cloud pressures that are too low compared to independent information [Boersma et al., 2011; Veefkind et al., 2016], so that more OMI pixels pass the selection criteria. The bias for DOMINO v2 is $+0.85 \times 10^{15}$ molec.cm$^{-2}$ (+11%, $n$=45), with a root mean squared deviation of $2.66 \times 10^{15}$ molec.cm$^{-2}$ (35%).

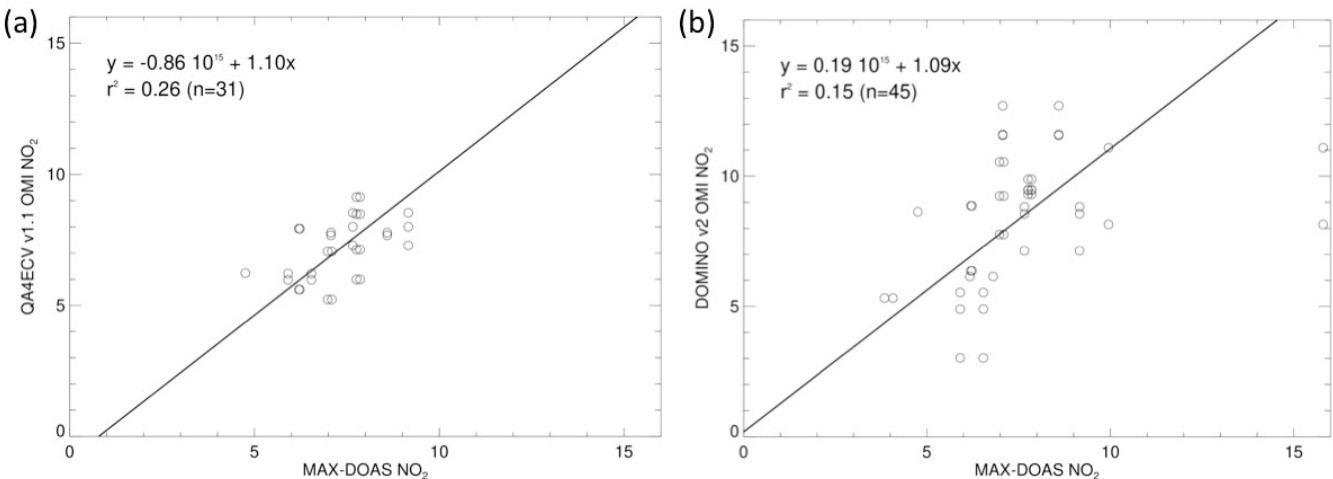

**Figure 13.** (a) Scatterplot of QA4ECV OMI vs. MAX-DOAS tropospheric NO$_2$ columns for Tai'an (China) in May-June 2006. The solid line shows the result of a reduced major axis regression to the data. Only pixels measured a cloud radiance fraction < 0.5 and an effective cloud pressure <850 hPa, within 20 km and 30 minutes of a MAX-DOAS measurement have been selected. (b) Same as (a), but

now for DOMINO v2 vs. MAX-DOAS tropospheric NO$_2$.





The differences between OMI and MAX-DOAS $NO_2$ columns provide an opportunity to evaluate the uncertainties of the satellite retrievals. This relies on good knowledge of the MAX-DOAS uncertainties and relatively small uncertainties associated with the representativeness of MAX-DOAS for the coincident OMI columns. Assuming that the retrieval errors between OMI and QA4ECV are independent and follow a normal distribution, we expect that the distribution of the differences between OMI and MAX-DOAS takes on a Gaussian form characterized by width $\sigma = (\sigma_O^2 + \sigma_{MD}^2 + \sigma_R^2)^{-1/2}$, with $\sigma_O$ the uncertainty reported (in the data files) for QA4ECV OMI $NO_2$ columns, $\sigma_{MD}$ the uncertainty reported for MAX-DOAS $NO_2$ columns, and $\sigma_R$ the uncertainty from spatiotemporal mismatches between the satellite and ground-based measurement (Table 7). The mean reported uncertainties ($\sigma_O$ and $\sigma_{MD}$) are regarded as random errors here (see discussion in Section 2.3 in Boersma et al. [2004], and below). In Figure 14 we compare the distribution of differences predicted from the above Gaussian function based on the uncertainties reported in the OMI and MAX-DOAS data files and a 10% representativeness difference error (estimated from deviations from the Tai'an value shown in Figure S1), to the actually observed differences (individual pairs of OMI and MAX-DOAS $NO_2$ column values). We see from Figure 14 that the differences between OMI and MAX-DOAS $NO_2$ columns are more narrowly distributed than expected from algorithm uncertainties and theory, although the sample size is small (n=31 for QA4ECV, n=45 for DOMINO v2). This holds for QA4ECV differences, which are 39% smaller than expected, but also for the DOMINO v2 differences, 16% smaller than expected over Tai'an. The tighter distribution of the observed differences implies that: (1) the uncertainties in OMI and MAX-DOAS retrievals possess some degree of correlation (for instance in situations when OMI is biased high, also MAX-DOAS may be biased high, limiting the magnitude of the differences), (2) OMI and/or MAX-DOAS algorithm uncertainty estimates are too conservative, (3) OMI or MAX-DOAS uncertainties contain an unknown persistent error component, so that $\sigma_O$ or $\sigma_{MD}$ have been overestimated, or (4) a combination of the above. The MAX-DOAS $NO_2$ retrieval technique suffers from some similar error contributions (a priori $NO_2$ profile shape, aerosols) but it is also different by design from the satellite retrieval (no albedo or stratospheric correction dependence, ground-based perspective), so we should expect some but not full error correlation. If there would be a substantial



systematic and persistent error component to $\sigma_O$ or $\sigma_{MD}$ (and we have no indication for this nor do we know about its magnitude or sign), we would have needed to reduce our estimates for $\sigma_O$ and $\sigma_{MD}$ in Table 7 and expect a distribution of the differences that is more narrowly Gaussian and peaking at a typical systematic difference (or bias). Figure 14 shows little bias for QA4ECV. We therefore conclude

5    that it is likely that the OMI (and MAX-DOAS) retrieval uncertainties estimates are too conservative, although our findings are based on a small sample. In the case of QA4ECV, a reduction of both the OMI and MAX-DOAS uncertainties by 35%, would be in much better agreement with the observed differences at the Tai'an station.

10   **Table 7.** Expected and observed differences between OMI and MAX-DOAS NO₂ columns observed over Tai'an in June 2006 for the QA4ECV (n=31) and DOMINO v2 (n=45) ensemble. Summary of uncertainties for the all (31 or 45) matching pixels. $\sigma_O$ (reported OMI uncertainty) and $\sigma_M$ (reported MAX-DOAS uncertainty) are the mean of 31 or 45 individual values, and $\sigma_R$ is considered to be a 10% contribution from mismatches.

|  | Expected differences (QA4ECV) | Observed differences (QA4ECV) | Expected differences (DOMINO v2) | Observed differences (DOMINO v2) |
|---|---|---|---|---|
| $\sigma$ | $2.11\times10^{15}$ molec. cm$^{-2}$ | $1.29\times10^{15}$ molec. cm$^{-2}$ | $2.57\times10^{15}$ molec. cm$^{-2}$ | $2.16\times10^{15}$ molec. cm$^{-2}$ |
| $\sigma_O$ | $1.71\times10^{15}$ molec. cm$^{-2}$ |  | $2.21\times10^{15}$ molec. cm$^{-2}$ |  |
| $\sigma_M$ | $1.08\times10^{15}$ molec. cm$^{-2}$ |  | $1.27\times10^{15}$ molec. cm$^{-2}$ |  |
| $\sigma_R$ | $0.60\times10^{15}$ molec. cm$^{-2}$ |  | $0.60\times10^{15}$ molec. cm$^{-2}$ |  |

This first validation is based on a limited time range and one site. A more comprehensive validation work, based on several MAXDOAS sites and several years of data is in preparation [Compernolle 2018].



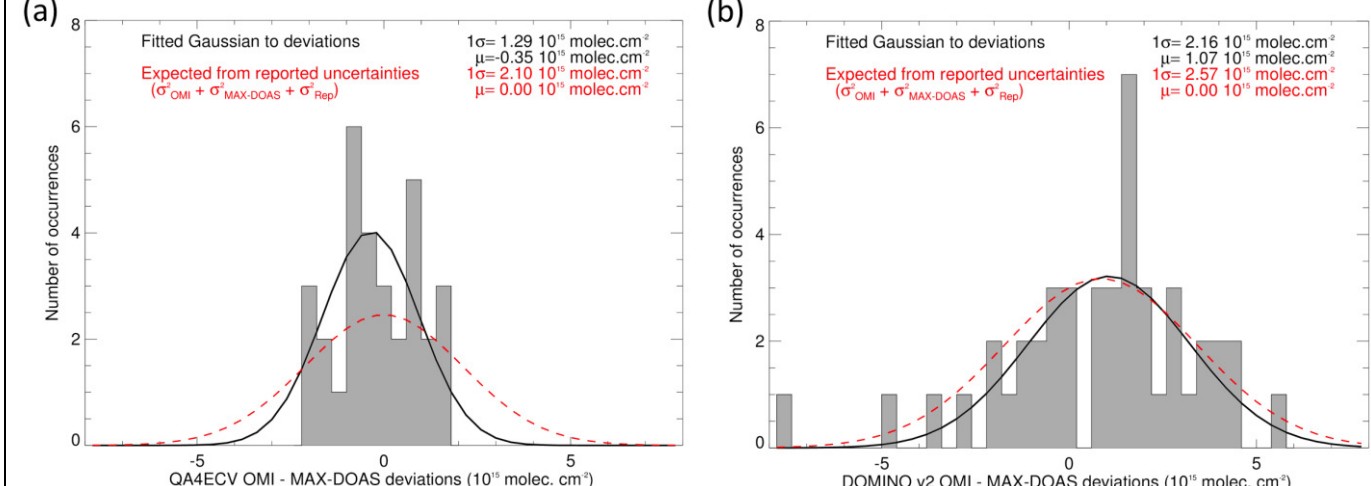

**Figure 14.** (a) Histogram of differences of QA4ECV OMI vs. MAX-DOAS tropospheric $NO_2$ columns for Tai'an (China) in May-June 2006. The black line shows a Gaussian fit to the observed differences, and the red dashed lines shows the Gaussian expected from the uncertainties reported in the QA4ECV and MAX-DOAS data products. (b) Same as (a), but now for differences between DOMINO v2 vs. MAX-DOAS tropospheric $NO_2$.

## 8 Data availability and DOI

The QA4ECV $NO_2$ Essential Climate Variable precursor product contains vertical $NO_2$ columns for the
5  period 1995-2017. The dataset contains: (1) the tropospheric vertical column density, (2) the stratospheric vertical column density, and (3) the total vertical column density. The $NO_2$ ECV precursor data provides geophysical information for each and every ground pixel observed by GOME, SCIAMACHY, OMI, and GOME-2(A). The QA4ECV $NO_2$ data product is available online via www.qa4ecv.eu, under 'ECV data'. The data product has been processed with the coherent algorithm
10  described in this work.

For GOME, data is available from 1 July 1995 until 30 June 2003 (8 years). For SCIAMACHY, data is available from 1 August 2002 until 30 April 2012 (9 years and 9 months). For GOME-2(A), data is available from 1 January 2007 until 31 December 2017, and for OMI from 1 October 2004 until 31



December 2017, so that the total length of the data set exceeds 22 years at the time of writing. For each of the data sets, digital object identifiers have been registered [Boersma et al., 2017[b,c,d,e]]. Detailed information on how to use the data can be found in the Product Specification Document for $NO_2$ ECV Precursor product [Boersma et al., 2017[f]].

## 9 Summary

We have developed an improved algorithm and uncertainty assessment for tropospheric $NO_2$ satellite retrievals from UV/VIS satellite sensors. Our effort has resulted in the generatation of a 1995-2017 climate data record of tropospheric $NO_2$ columns with fully traceable uncertainty metrics that can be

readily used for model evaluation, for estimating $NO_x$ emissions and nitrogen deposition. In designing our new algorithm, we followed advice from the user and producer community and from WMO GCOS best practices on generating climate data records. Specifically, we extended the information content on flags and uncertainties in the data files, and present a so-called traceability chain along with the data files. This traceability chain is an easily accessible web-hosted interactive flow diagram that shows the

components of the QA4ECV $NO_2$ algorithm, and how external information is embedded in the retrieval process, providing details on where those pieces of information can be found.

The QA4ECV project involved detailed comparisons of different approaches between groups for the DOAS slant column retrievals and the estimate of the stratospheric sub-column and air mass factors.

Using the latest and best available level-1 data for GOME, SCIAMACHY, GOME-2A, and OMI from the relevant space agencies, the comparisons led us to improve the spectral fitting of $NO_2$ by accounting for liquid water absorption, and an intensity-offset correction. This improved the quality of the $NO_2$ fit over clear-sky ocean scenes by up to 30% [Zara et al., 2018], but did not substantially affect the $NO_2$ fits over polluted scenes. We compared three alternate methods for estimating the stratospheric $NO_2$

background: data assimilation [Dirksen et al., 2011], and the model-based [Hilboll et al., 2013] and modified reference sector (STREAM, Beirle et al., [2016]) approaches. Data assimilation was considered to be the most viable option for the QA4ECV algorithm because it provides a coherent framework for stratospheric corrections as well as air mass factor (AMF) calculations. We based the





data assimilation on the TM5-MP chemistry transport model with 1°×1° horizontal resolution, a major step forward compared to earlier assimilation schemes based on TM4 (3°×2°), and include corrections for sphericity effects on atmospheric radiative transfer, as described in Lorente et al. [2017]. Our new stratospheric correction leads to fewer negative tropospheric $NO_2$ columns for retrievals at extreme

viewing geometries. We then tested various models and approaches to calculate tropospheric AMFs under challenging retrieval scenarios. AMFs calculated with different radiative transfer models agree well, as long as assumptions and ancillary data inputs are consistent. With groups using their own preferred settings, we find differences (or structural uncertainty) in AMFs up to 40% with respect to the ensemble mean, stressing the importance of adequate traceability. Many of the lessons learned for

QA4ECV algorithm development, are currently being applied to $NO_2$ retrievals from S5P-TROPOMI.

The QA4ECV $NO_2$ product contains an algorithm uncertainty estimate associated with each individual observation. We obtain this estimate via uncertainty propagation calculations, accounting for pixel-specific sensitivities to state parameters (Jacobians) such as surface reflectance, clouds, and the $NO_2$

vertical profile. The uncertainties are highest in the cold season, when AMFs are particularly uncertain, and typically amount to 40% over polluted area. For averaged QA4ECV $NO_2$ data, associated uncertainties may be reduced, but part of the uncertainty due to systematic error will remain. Our work provides recommendations on how to estimate the uncertainty for spatially or temporally averaged data, taking into account a partial correlation in the errors between pixels. We evaluated the algorithm

uncertainties against independent assessments of structural uncertainties for each retrieval step and find that the structural uncertainties are of similar magnitude or exceed the algorithm uncertainties for all retrieval sub-steps. Finally, we used MAX-DOAS $NO_2$ column measurements obtained over the polluted Tai'an (China) region in June 2006 to validate the OMI QA4ECV $NO_2$ columns and their uncertainties. Accounting for spatial differences between the pixel and the location of Tai'an, we found

good agreement between the QA4ECV and MAX-DOAS $NO_2$ columns (bias = -2%, RMS differences 13%, $n$=31), much better than the agreement between DOMINO v2 and MAX-DOAS (bias = +11%, RMS 28%, $n$=45). The small differences between coinciding QA4ECV and MAX-DOAS $NO_2$ columns





suggest that our QA4ECV algorithm uncertainties are likely on the conservative side, at least over Tai'an.

**Acknowledgments**

This research has been supported by the EU FP7 Project Quality Assurance for Essential Climate Variables (QA4ECV), grant no. 607405. We acknowledge the free use of the GOME-2 data provided by EUMETSAT. We thank all the persons who have kindly and carefully responded to QA4ECV user survey. We appreciate the efforts by Michael Barkley (Leicester University), Lok Lamsal (NASA GSFC), Jin-Tai Lin and Mengyao Liu (Peking University) who kindly contributed to the $NO_2$ AMF
intercomparison.

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
