# Peer review of "Improving algorithms and uncertainty estimates for satellite $NO_2$ retrievals: Results from the Quality Assurance for Essential Climate Variables (QA4ECV) project"

_Atmospheric Measurement Techniques, 2018_

## Referee Comment (RC1) · Anonymous Referee #1 · 7 Sep 2018

Improving algorithms and uncertainty estimates for satellite NO 2 retrievals: Results from the Quality Assurance for Essential Climate Variables (QA4ECV) project, by K. F. Boersma et al.

MS No.: amt-2018-200

The manuscript summarizes the long-term effort in building a consistent, multi- decade record of NO2 columns based on the practically all available data sets acquired from space. The article offers a plenitude of technical details aiming to improve the traceability of the described products as well as provides valuable recommendations that should help improving quality of the NO2 retrievals. The article deserves a prompt publication.

I would like the authors to consider the following corrections/amendments:

=================================================================
The main text:

I think that Section 2, especially Sect. 2.1, should be substantially shortened, since, by the author's remark, the survey's outcome was published elsewhere. I am not sure Fig. 1 adds any valuable content to the main objectives of the article. I would consider its removal. Section 2.1 could be shortened to 1-2 paragraphs by mentioning only the user's suggestions that have been implemented in the current version of the products). I would leave out all 'things-to-be-done' or 'things-to-be-considered'.

The same applies to Sect. 2.2 - I would concentrate exclusively on the already implemented items.

If the authors perceive the full review of the user's comments as very important, then I suggest moving it to Appendix.

Also, consider putting the text from p.8, l.25 through p.10, l.21 into a new Sect. 2.3, thus ascribing the current Sect. 2.3 to 2.4. This particular text, [p.8,l.25 - p.10,l.21], does not belong in Sect. 2.2.

The Footnote #2 to Table 1 does not help understand the meaning of the quoted 2%. Please either re-phrase or remove.

Table 2: is GOME v5 L1 used in the article different from the most recent L1 version described by Coldewey-Egbers et al. (AMTD, 2018)? Adding a footnote may help to remove the ambiguity. Also, Shah et al. (2018, AMT, 11, 2345) describes SCIAMACHY V8 while the authors use v7. This should be commented on just as well.

[Figure]

p.15, l.5. OMI shows different degradation rates in radiances and irradiances. More-over, various instrument-performance metrics show far superior stability than the quoted 2%. The wording should be changed to something like: "The OMI instrument produces stable (to ∼2% over the mission time, in the row anomaly-free areas) L1B radiances".

p.15, l.11. '... directly from the Sun...'. Please re-phrase, since 'directly' does not apply to the sunlight presumably scattered by a peeling piece of insulation.

Fig. 2. The percentages shown at the upper axis do not correspond to the RA-marked areas in the plot. Either correct or clarify.

p.16, l.1. Replace 'detector degradation' by 'optical throughput changes in the irradi-ance channel'. The CCD detector does not change at the quoted rates.

p.18, l.9. In addition to the mentioned time-dependent slit-function changes, the GOME-2A slit function varies with the scan angle and along the orbit, though at a much lower level compared to the dominant long-term changes. Are these cross-track and along-orbit changes accounted for? Please comment on.

Figure 3. In the present format some details are inevitably lost due to the limited reso-lution of a printout. The same applies to the screen viewing, no matter the zoom. I may suggest: 1. Substantially enlarging the upper section. 2. Substantially overlapping two lower sections (these are practically the same) and expanding both of them at the same time.

p.23, l.4. Do you include or reject the RA-affected retrievals in your stats? Please clarify.

Fig. 4. Define the units of the color bars under both plots.

p.23, l.13. The intensity offset proves to be very important, so this particular subject deserves more detailed discussion, either in the main text or in a separate Appendix. In particular, the authors should provide a mathematical description and discuss the

particulars of the 'best-practice' implementation of the offset. Indeed, the cited Müller et al. [2016] provides valuable information, but it is not possible to conclude which form of the intensity offset is considered as the best-practice by the authors. I suggest, besides providing mathematical description in the text, putting some specifics of the applied approach in Table 3. Currently the Table carries 'yes' or 'no' entries in the respective column. Does this mean that all 'yes' entries ascribe to the same intensity-offset mathematical form and implementation?

Table 4. Describe the 'undersampling' and 'Eta' corrections. These weren't mentioned in the text.

p.31, Footnote 6 : replace '...behavior of the red line...' by '...behavior of the black line...'

p.32, l.1. The 1st sentence mentions AMF_strat in Fig.5, while Fig.5 caption says AMF_total. Which one is true?

Table 5. If all listed values are derived on a pixel basis, then I would put a note in the caption and remove 'per pixel' from the fields. If not, then each field must carry either 'global' or 'per pixel' notation. Or, better yet, the table could be segregated into the 'pixel' and 'global' parts.

Fig. 9. All the colors but blue and light-blue are described in the caption. Either add these two descriptions or remove all and mention that additional info is provided in the figure legends.

Fig.13. The caption quotes P < 850 hPa while the text (p. 50) says P < 875 hPa. Which one applies?

p.52, l.7. The common Gauss-process error-combination formula gives 'sigma=sqrt(sum(sigma_i ^2))'. The text mentions 'sigma = 1 / sqrt(sum(sigma_i ^2))'. Please explain the difference.

Summary:

p.50 quotes -2% bias and 16% RMS, while Summary provides -2% and 13%, respectively. And, the same applies to DOMINO results: [+11% and 35%] in p.51 vs. [+11% and 28%] in Summary. Please clarify.

The list of references: a quick spot-check shows the cited in the text, but missing in the list - Boersma et al. [2004], Irie et al. [2009], Krotkov et al. [2016]. Thus, I suspect such omissions to be more numerable. Please double-check the list of references.

Also, please add letters b through f to the Boersma et al. [2017] works, to match the citations in the main text.

==================================================================

Additional corrections:

p.23, l.15 - "... results in larger NO2..."

p.24, l.10 - "The inter-comparison of preferred-setting SCDs..."

p.29, l.4 - remove the 2nd 'agree reasonably well'

---

## Referee Comment (RC2) · Anonymous Referee #2 · 27 Sep 2018

The paper "Improving algorithms and uncertainty estimates for satellite NO2 retrievals: Results from the Quality Assurance for Essential Climate Variables (QA4ECV) project" by Boersma et al. describes the improvement of NO2 retrieval algorithms for GOME, SCIAMACHY, OMI and GOME-2, and the production of a multi-decadal record of global NO2 columns. This manuscript provides an overview of the NO2 retrieval results from the QA4ECV project, which have been partially reported in earlier papers. Furthermore, first comparisons have been made between the OMI tropospheric NO2 data and ground-based MAX-DOAS measurements at one site in China.

[Figure]

The topic of the manuscript is within the scope of AMT and it is of interest to the scientific community. It can be recommended for publication, if the authors make an effort to address the comments listed below, and improve the manuscript accordingly.

Specific comments:

Section 3

Table 2 The spatial resolution of GOME-2A was changed to 40x40 km2 on 15 July 2013.

P18 The diffuser plate also causes spectral structures in the GOME-2 solar irradiances, but the effect is much smaller than for GOME/ERS-2.

Section 5.1

The manuscript discusses a consistent NO2 retrieval for GOME, SCIAMACHY, OMI and GOME-2, and Table 4 lists the specific DOAS settings for the 4 satellite instruments. However, in the text only the analyses of the fitting results for OMI are discussed in detail. Considering the specific instrument characteristics affecting the DOAS retrieval (as described in Sect. 3), a short discussion on the NO2 fitting results for the other instruments should be included, and the consistency between the results of the 4 instruments discussed.

P23-24 The differences in the NO2 slant column shown in Fig. 4 are significant. As described in the text, the differences are mainly a result of the intensity vs optical density fit and the intensity offset (while the stratospheric AMF effect is very small). These two key DOAS subjects should be discussed in more detail in this section. In particular, I suggest adding a discussion/recommendation about the intensity vs optical density fit method.

Fig 5b The illustrated scenario (SZA=VZA=0) is not really a typical OMI viewing geometry. Please use a realistic mid-latitude OMI measurement scenario.

Section 5.2

P27 Here it is mentioned that NO2 SCDs are assimilated in TM4, while at the end of Sect. 5.3 TM5 is mentioned.

P28 Please provide examples of "particular applications" of the two different STS approaches.

Section 6.2

Tabel 5 Are the "Symbols" for the three "Contribution to the uncertainty of . . ." entries correct?

Section 7

As also mentioned in the text, the ground-based validation discussed in the paper is for one site only and has a very limited time range. Although it is understandable that in this manuscript only a first validation is presented and a dedicated validation paper is in preparation, it is problematic to draw conclusions about OMI retrieval uncertainties based on regression/differences analysis of such a small data sample. If the validation and uncertainty analysis are illustrated for one site then a longer data period should be used to increase the number of measurements and to be able to account for seasonal variations. Why has the site Tai'an been selected if only campaign data for a short period is available? (also considering the fact that the authors have access to longer MAX DOAS data records for several other sites).

---

## Author Comment (AC1) · 8 Nov 2018

We thank the reviewer for the positive and useful comments, and for their careful reading of the paper. We have addressed the questions as follows, with our response in blue.

**Reviewer Comment 2**

The paper "Improving algorithms and uncertainty estimates for satellite NO2 retrievals: Results from the Quality Assurance for Essential Climate Variables (QA4ECV) project"

[Figure]

by Boersma et al. describes the improvement of NO2 retrieval algorithms for GOME, SCIAMACHY, OMI and GOME-2, and the production of a multi-decadal record of global NO2 columns. This manuscript provides an overview of the NO2 retrieval results from the QA4ECV project, which have been partially reported in earlier papers. Furthermore, first comparisons have been made between the OMI tropospheric NO2 data and ground-based MAX-DOAS measurements at one site in China. The topic of the manuscript is within the scope of AMT and it is of interest to the scientific community. It can be recommended for publication, if the authors make an effort to address the comments listed below, and improve the manuscript accordingly.

Thank you for this comment.

Section 3

Table 2 The spatial resolution of GOME-2A was changed to 40x40 km$^2$ on 15 July 2013.

We have now included this in Table 2.

P18 The diffuser plate also causes spectral structures in the GOME-2 solar irradiances, but the effect is much smaller than for GOME/ERS-2.

We have now added this sentence to Section 3.

Section 5.1

The manuscript discusses a consistent NO2 retrieval for GOME, SCIAMACHY, OMI and GOME-2, and Table 4 lists the specific DOAS settings for the 4 satellite instruments. However, in the text only the analyses of the fitting results for OMI are discussed in detail. Considering the specific instrument characteristics affecting the DOAS retrieval (as described in Sect. 3), a short discussion on the NO2 fitting results for the other instruments should be included, and the consistency between the results of the 4 instruments discussed.

[Figure]

Good point. Although the focus within QA4ECV was initially on OMI, we also evaluated the consistency in the NO2 SCDs from the other instruments. We have now added a discussion of our tests on the consistency of the NO2 fitting results from GOME-2, SCIAMACHY, and GOME.

P23-24 The differences in the NO2 slant column shown in Fig. 4 are significant. As described in the text, the differences are mainly a result of the intensity vs optical density fit and the intensity offset (while the stratospheric AMF effect is very small). These two key DOAS subjects should be discussed in more detail in this section. In particular, I suggest adding a discussion/recommendation about the intensity vs optical density fit method.

We discuss in more detail now the use of the optical density fits and intensity offset used in the QA4ECV fitting process. For a detailed discussion of the tests done, we refer to QA4ECV D4.2, and a work in preparation on these test by the University of Bremen.

Fig 5b The illustrated scenario (SZA=VZA=0) is not really a typical OMI viewing geometry. Please use a realistic mid-latitude OMI measurement scenario.

We have done this, and now show the scenario for SZA = VZA = 30°.

Section 5.2 P27 Here it is mentioned that NO2 SCDs are assimilated in TM4, while at the end of Sect. 5.3 TM5 is mentioned.

In the testing phase, we used the DOMINO v2 code, where assimilation is done in TM4 [Boersma et al., 2011; Dirksen et al., 2011]. In the retrieval development phase, we updated the model framework to TM5. The text now reflects this.

P28 Please provide examples of "particular applications" of the two different STS approaches.

The purpose of either STS approach is to correct for the stratospheric contribution to the observed SCD, in order to retrieve a high-quality tropospheric product. Both

corrections perform similarly over regions with strong anthropogenic NO2 pollution.

Data assimilation can be applied for global retrievals. The strength of data assimilation is that it provides a correction that is based on broad and consistent knowledge on the state of the atmosphere (NO2 and temperature profile, stratospheric dynamics). Especially in situations with strong stratospheric NO2 gradients, such as near the polar vortex, assimilation is the preferred approach. It has been shown that the data assimilation captures the strong spatial gradients occurring near the vortex [Dirksen et al., 2011], whereas the STREAM method by design results in zonally smooth structures in those regions.

STREAM can be applied especially for studies into small sources, such as emissions from soil, ships, and small, isolated anthropogenic sources. The strength of STREAM is that it does not rely on chemistry transport models. Data assimilation is potentially somewhat vulnerable to misinterpret tropospheric contributions as stratospheric NO2. This qualifies STREAM as a useful alternative for data assimilation for studies in areas away from strong stratospheric gradients (where the zonally smooth structure of the stratospheric field is of little consequence). The text has been updated accordingly.

Section 6.2 Tabel 5 Are the "Symbols" for the three "Contribution to the uncertainty of . . ." entries correct?

They are indeed correct, but could be clarified a bit more. The first term, between brackets, indicates the AMF uncertainty caused by uncertainty in the forward model parameters. These terms are the same as Eq. (12) in Boersma et al. [2004]. Since we're interested here in the contribution of these uncertainties to the NO2 column, we ratio the AMF uncertainty by the AMF itself, and multiply with the actual tropospheric NO2 column. We now include a clarification in a footnote to Table 5.

Section 7 As also mentioned in the text, the ground-based validation discussed in the paper is for one site only and has a very limited time range. Although it is understandable that in this manuscript only a first validation is presented and a dedicated validation

paper is in preparation, it is problematic to draw conclusions about OMI retrieval uncertainties based on regression/differences analysis of such a small data sample. If the validation and uncertainty analysis are illustrated for one site then a longer data period should be used to increase the number of measurements and to be able to account for seasonal variations. Why has the site Tai'an been selected if only campaign data for a short period is available? (also considering the fact that the authors have access to longer MAX DOAS data records for several other sites).

A more comprehensive validation work, with focus on other sites and over longer periods, is in preparation by our Belgian colleagues (as was mentioned in the manuscript). We chose to do a first evaluation of the OMI QA4ECV NO2 product and their uncertainties with the Tai'an data available to us, because these data have been used before to validate the DOMINO v2 product with good agreement [Irie et al., 2012]. This anchors our evaluation of the new QA4ECV product to previous, well-established efforts. Moreover, the scope in section 7 is not just on validation but as much on bringing forward ideas on evaluating satellite data, such as correcting for spatial gradients, and using the deviations to check on the reported uncertainties.

---

## Author Comment (AC2) · 9 Nov 2018

We thank the reviewer for the positive and useful comments, and for careful reading of the paper. We have addressed the questions as follows, with our response in blue.

**Reviewer Comment 1**

The manuscript summarizes the long-term effort in building a consistent, multi- decade record of NO2 columns based on the practically all available data sets acquired from space. The article offers a plenitude of technical details aiming to improve the traceability of the described products as well as provides valuable recommendations that should help improving quality of the NO2 retrievals. The article deserves a prompt publication.

Thank you for this comment.

I would like the authors to consider the following corrections/amendments: ================================================================
The main text: I think that Section 2, especially Sect. 2.1, should be substantially shortened, since, by the author's remark, the survey's outcome was published elsewhere. I am not sure Fig. 1 adds any valuable content to the main objectives of the article. I would consider its removal. Section 2.1 could be shortened to 1-2 paragraphs by mentioning only the user's suggestions that have been implemented in the current version of the products). I would leave out all 'things-to-be-done' or 'things-to-be-considered'.

The same applies to Sect. 2.2 - I would concentrate exclusively on the already implemented items.

If the authors perceive the full review of the user's comments as very important, then I suggest moving it to Appendix.

This is a good point. We feel it is important to document in an easily accessible manner the outcomes of the surveys, and this is now done in the Supplementary material. We shortened the text and removed original Figure 1 accordingly.

Also, consider putting the text from p.8, l.25 through p.10, l.21 into a new Sect. 2.3, thus ascribing the current Sect. 2.3 to 2.4. This particular text, [p.8,l.25 - p.10,l.21], does not belong in Sect. 2.2.

Agreed. We now introduce a new section 2.3 titled 'QA4ECV consortium activities'.

The Footnote 2 to Table 1 does not help understand the meaning of the quoted 2

[Figure]

We have rephrased this as follows: "According to GCOS, the user requirement for stability is a requirement on the extent to which the uncertainty of a measurement remains constant over a long period (GCOS-200, 2016)."

Table 2: is GOME v5 L1 used in the article different from the most recent L1 version described by Coldewey-Egbers et al. (AMTD, 2018)? Adding a footnote may help to remove the ambiguity. Also, Shah et al. (2018, AMT, 11, 2345) describes SCIAMACHY V8 while the authors use v7. This should be commented on just as well.

Thanks for the opportunity to clarify this. In our GOME data product, we worked with what was available to us at the time of project finalization, which was v5 level 1 data. This is not exactly the same as in Coldewey-Egbers et al. [2018], which is v5.1. The main difference between v5 and v5.1 is the consistency of orbits, and not the radiances themselves [Dehn, personal communication], which is not a strong concern.

For SCIA, we indeed use V8 level 1 data, so our original manuscript wrongfully stated that it was v7.04-w. This has now been corrected.

p.15, l.5. OMI shows different degradation rates in radiances and irradiances. Moreover, various instrument-performance metrics show far superior stability than the quoted 2

Done.

p.15, l.11. '... directly from the Sun...'. Please re-phrase, since 'directly' does not apply to the sunlight presumably scattered by a peeling piece of insulation.

Done.

Fig. 2. The percentages shown at the upper axis do not correspond to the RA-marked areas in the plot. Either correct or clarify.

The percentages in the upper x-axis do not correspond to (#badrow/#allrows)x100% but to the "percentage of affected pixels" meaning: (#badpixels/#totalpixels)x100%

from all rows. For some rows, the row anomaly does not affect the entire orbit. This is now clarified in the caption of the figure concerned.

p.16, l.1. Replace 'detector degradation' by 'optical throughput changes in the irradiance channel'. The CCD detector does not change at the quoted rates.

Agreed. We updated the sentence.

p.18, l.9. In addition to the mentioned time-dependent slit-function changes, the GOME-2A slit function varies with the scan angle and along the orbit, though at a much lower level compared to the dominant long-term changes. Are these cross-track and along-orbit changes accounted for? Please comment on.

In the GOME-2 NO2 fits, no slit function fit was performed so in-orbit changes are not accounted for. In the visible, the change of the GOME-2(A) slit function is not important (unlike in the UV-range). If any cross-track changes in the slant columns would still persist, these are likely dampened by the stripe-correction, which was applied on GOME-2A data in the same fashion as for OMI.

Figure 3. In the present format some details are inevitably lost due to the limited resolution of a printout. The same applies to the screen viewing, no matter the zoom. I may suggest: 1. Substantially enlarging the upper section. 2. Substantially overlapping two lower sections (these are practically the same) and expanding both of them at the same time.

Thanks for this good suggestion. We followed up on it.

p.23, l.4. Do you include or reject the RA-affected retrievals in your stats? Please clarify.

We have applied XTrack flagging using the mask provided in the lv1 files, which means that the RA-affected retrievals have not been included in the stats.

Fig. 4. Define the units of the color bars under both plots.

Done.

p.23, l.13. The intensity offset proves to be very important, so this particular subject deserves more detailed discussion, either in the main text or in a separate Appendix. In particular, the authors should provide a mathematical description and discuss the particulars of the 'best-practice' implementation of the offset. Indeed, the cited Müller et al. [2016] provides valuable information, but it is not possible to conclude which form of the intensity offset is considered as the best-practice by the authors. I suggest, besides providing mathematical description in the text, putting some specifics of the applied approach in Table 3. Currently the Table carries 'yes' or 'no' entries in the respective column. Does this mean that all 'yes' entries ascribe to the same intensity offset mathematical form and implementation?

We now refer to the mathematical description of the intensity offset which has been applied. The selected form of the intensity offset was driven by the optical density nature of the fit.

Table 4. Describe the 'undersampling' and 'Eta' corrections. These weren't mentioned in the text.

Good point. We now elaborate on these corrections in the margin of Table 4.

p.31, Footnote 6 : replace '...behavior of the red line...' by '...behavior of the black line...'

Done.

p.32, l.1. The 1st sentence mentions AMF_strat in Fig.5, while Fig.5 caption says AMF_total. Which one is true?

Thank you for pointing this out. We corrected the first sentence on page 32.

Table 5. If all listed values are derived on a pixel basis, then I would put a note in the caption and remove 'per pixel' from the fields. If not, then each field must carry either 'global' or 'per pixel' notation. Or, better yet, the table could be segregated into the

'pixel' and 'global' parts.

Thank you for this valuable suggestion. We followed up on this suggestion and introduced a category indicating whether the estimate is pixel-specific, 'global', or a mix of these categories.

Fig. 9. All the colors but blue and light-blue are described in the caption. Either add these two descriptions or remove all and mention that additional info is provided in the figure legends.

Done.

Fig.13. The caption quotes P < 850 hPa while the text (p. 50) says P < 875 hPa. Which one applies?

875 hPa applies. Corrected.

p.52, l.7. The common Gauss-process error-combination formula gives 'sigma=sqrt(sum(sigma_$i^2$))'. The text mentions 'sigma = 1 / sqrt(sum(sigma_$i^2$))'. Please explain the difference.

This was corrected and now reads 'sigma=sqrt(sum(sigma_$i^2$))'.

p.50 quotes -2% bias and 16% RMS, while Summary provides -2% and 13%, respectively. And, the same applies to DOMINO results: [+11% and 35%] in p.51 vs. [+11% and 28%] in Summary. Please clarify.

Corrected. The numbers in the text apply, and those in the summary has been corrected.

The list of references: a quick spot-check shows the cited in the text, but missing in the list - Boersma et al. [2004], Irie et al. [2009], Krotkov et al. [2016]. Thus, I suspect such omissions to be more numerable. Please double-check the list of references.

Done.

Also, please add letters b through f to the Boersma et al. [2017] works, to match the citations in the main text.

Done.

===================================================================
Additional corrections: p.23, l.15 - "... results in larger NO2..."

Done.

p.24, l.10 - "The inter-comparison of preferred-setting SCDs..."

Done.

p.29, l.4 - remove the 2nd 'agree reasonably well'

Done.